# Loss of Aspm causes increased apoptosis of developing neural cells during mouse cerebral corticogenesis

**Madoka Tonosaki[1], Akira Fujimori[2], Takeshi Yaoi[1], Kyoko Itoh[1] ***

**1** Department of Pathology and Applied Neurobiology, Kyoto Prefectural University of Medicine, Graduate School of Medical Science, Kyoto, Japan, **2** Department of Basic Medical Sciences for Radiation Damages, National Institute of Radiological Sciences (NIRS), National Institutes for Quantum and Radiological Science and Technology (QST), Chiba, Japan

* kxi14@koto.kpu-m.ac.jp

**Data Availability Statement:** All relevant data are within the paper and its Supporting Information files. Data Availablity Statement The data that support the findings of this study are openly available in figshare at 10.6084/m9.figshare.

## Abstract

Abnormal spindle-like microcephaly associated (ASPM) is a causative gene of primary autosomal recessive microcephaly. Microcephaly is considered to be a consequence of a small brain, but the associated molecular mechanisms are not fully understood. In this study, we generated brain-specific Aspm knockout mice to evaluate the fetal brain phenotype and observed cortical reduction in the late stage of murine cortical development. It has been reported that the total number of neurons is regulated by the number of neural stem and progenitor cells. In the Aspm knockout mice, no apparent change was shown in the neural progenitor cell proliferation and there was no obvious effect on the number of newly generated neurons in the developing cortex. On the other hand, the knockout mice showed a constant increase in apoptosis in the cerebral cortex from the early through the late stages of cortical development. Furthermore, apoptosis occurred in the neural progenitor cells associated with DNA damage. Overall, these results suggest that apoptosis of the neural progenitor cells is involved in the thinning of the mouse cerebral cortex, due to the loss of the Aspm gene in neocortical development.

## Introduction

Autosomal recessive primary microcephaly (MCPH) is a neurodevelopmental disorder, which is characterized by microcephaly, present at birth, and nonprogressive mental retardation. Microcephaly is the consequence of a small, but architecturally normal brain, and it is the cerebral cortex that shows the greatest size reduction. There are at least seven MCPH loci, and abnormal spindle-like microcephaly associated (ASPM) gene has been identified as the most common cause of MCPH: the MCPH5 gene [1–3]. It has been proposed that ASPM is a major determinant of cerebral cortical size among primates, including humans [4–6]. The function of the Aspm protein in neural progenitor cell expansion, as well as its localization to the mitotic spindle and midbody [7,8], suggest that it regulates brain development by a cellular division-related mechanism. In addition, the Aspm protein also causes a massive loss of germ

23626551 (Fig 1), 10.6084/m9.figshare.23626554 (Fig 2), 10.6084/m9.figshare.23626557 (Fig 3), 10.6084/m9.figshare.23626560 (Fig 4), 10.6084/m9.figshare.23626563 (Fig 5) 10.6084/m9.figshare.23626536 (S1 Fig), 10.6084/m9.figshare.23626515 (S2 Fig), 10.6084/m9.figshare.23626548 (S3), 10.6084/m9.figshare.23626539 (S4), 10.6084/m9.figshare.23626545 (S5 Fig), 10.6084/m9.figshare.23626527 (S6 Fig), 10.6084/m9.figshare.23626530 (S7 Fig), 10.6084/m9.figshare.23626533 (S8 Fig).

**Funding:** This work was supported by JSPS KAKENHI Grant Numbers JP25293240 (Kyoko Itoh), JP16K19689 and JP18K15683 (Madoka Tonosaki). There was no additional external funding received for this study. The funders had no role in study design, data collection and analysis, decision to publish, or preparation of the manuscript.

**Competing interests:** The authors have declared that no competing interests exist.

cells, resulting in a severe reduction in testis [9] and ovary size [10], accompanied by reduced fertility.

ASPM participates in the spindle organization, spindle positioning and cytokinesis process in all dividing cells. Aspm accumulates at the spindle poles during the M phase (metaphase) and regulates cell proliferation [7,11]. During neurogenesis, the majority of neurons and glia in the mammalian neocortex arise from the division of neural progenitor cells (NPCs) in the ventricular zone of the brain. Primary NPCs have a specific pattern of mitotic activity, that is, symmetrical and asymmetrical division. Each symmetrical division increases the progenitor cell number by generating two progenitor cells per division. Asymmetric neurogenic divisions produce one neuron and one progenitor cell. In the developing mammalian cortex, the division fate of a cell appears dependent upon the orientation of the mitotic spindle and hence the position of the cleavage furrow with respect to the apical surface of the neuroepithelium. Therefore, mutations in the ASPM gene are a cause of mitotic aberrations during neurogenesis. It has been shown that the loss of Aspm causes abnormal mitosis, leading to a prolonged cell cycle [12] and increased immature neuronal differentiation associated with abnormal symmetric division [13] in the neural stem and progenitor cells in mice during cerebral cortical formation.

At present, it is unknown whether or not the loss of Aspm causes apoptosis of NPCs or neurons in cortical development. In one report, the expression of Aspm was shown to be downregulated in the ventricular zone of irradiated fetal mouse brain, as well as in irradiated neurosphere cultures [14], which suggested that the suppression of Aspm by ionizing radiation could be the initial molecular target leading to a sequential microcephaly formation. When DNA damage induced by DNA double-strand breaks is too severe to repair normally, cells will undergo apoptosis. It has been reported that, in cultured cells, the suppression of Aspm increases DNA double-strand breaks under radiation exposure [15]. However, it is unclear whether or not Aspm deficiency increases apoptosis in brain tissue in normal environments.

In the present study, we showed that the loss of Aspm causes increased apoptosis, occurring most frequently in the early stages (E12.5) in NPCs and neurons, associated with DNA damage during brain formation, using multiple indicators of apoptosis (activation of caspases (cleaved caspase-3 (CC3)-positive cells), DNA fragmentation (TUNEL-positive cells), and nuclear condensation).

## Materials and methods

### Animals

All of the animal experiments conducted in this study were approved by the Institutional Review Board for Biomedical Research using Animals at Kyoto Prefectural University of Medicine, and the animals were handled according to the Institutional Guidelines and Regulations. In order to obtain NesCre;Aspm$^{flox/+}$ mice, Nestin-cre (C57BL/6-Tg(Nes-Cre)1Kag, Center for Animal Resources and Development (CARD), CARD-ID 650) mice [16] were mated with Aspm$^{flox/flox}$ mice [9]. Brain specific Aspm knockout mice (NesCre;*Aspm*$^{flox/flox}$, Aspm cKO) were generated by mating Aspm$^{flox/flox}$ mice and NesCre;*Aspm*$^{flox/+}$ mice and detected by genotyping.

### Genotyping

Mouse genotyping DNA was isolated from tail tissue followed by digestion at 95°C in 180 μL 50 mM NaOH for 10 min by heat inactivation, mixed with 20 μL 1 M Tris-HCl pH 8.0 and the supernatant was used for PCR. PCR was performed using the primer pairs to detect floxed Aspm and wild-type alleles were detected by PCR amplification using primer set 1 (Forward;

5'-CAGGAGGAGTACATTGATGAAACT-3' and Reverse; 5'- TGTAAGGGTTTGGACTGGATAT TT-3'). These primers amplify a 165 bp wildtype band and a 199 bp targeted (floxed Aspm) band. The Cre alleles were also detected by PCR amplification using the primer set 2 (Forward; 5'-TCGATGCAACGAGTGATGAG-3' and Reverse; 5'-TCGGCTATACGTAACAGGG-3') that amplified a 480 bp band.

The PCR conditions employed were as follows. After DNA denaturation at 94˚C for 5 s, there were 38 cycles at 98˚C for 5 s, followed by 55˚C for 5s, and 72˚C for 11 s (SapphireAmp-Fast PCR Master Mix (TakaraBio)).

### EdU and BrdU labeling

For the pulse-chase experiments, 5-Bromo-2'-deoxyuridine (BrdU dissolved in PBS, 1 mg/mL, 500 μL/mouse) or 5-ethynyl-2'-deoxyuridine (EdU dissolved in PBS, 500 μg/mL, 500 μL/ mouse) was injected intraperitoneally and chased for the number of times specified.

### Histological and immunofluorescence analyses

For the preparation of the cryosections, the embryonic brains were extracted from the dams after deep anesthesia induced by intraperitoneal injection of pentobarbital solution (pentobarbitaol sodium (Kyoritsu Seiyaku, Japan)/saline, 100 mg/20 mL/1 kg body weight), fixed with 4% paraformaldehyde at 4˚C for 6 hours, washed with PBS 3 times every hour and cryoprotected with 30% sucrose in PBS at 4˚C overnight. The brains were then embedded in O.C.T. compound (Sakura Finetek Japan Co., Ltd.), frozen in dry ice powder and stored at -80˚C. Cryosections were prepared using a cryomicrotome (LEICA CM1850, Leica, Japan) at 16 μm- (E12.5) or 10 μm-thickness (E14.5 and E16.5). Immunofluorescence staining of the cryosections was performed using the following primary antibodies; anti-Tbr1 (rabbit polyclonal, 1:500, ab31940, Abcam, Cambridge, UK), anti-Ctip2 (rat monoclonal, 1:500, ab18465, Abcam), anti-Satb2 (mouse monoclonal, 1:50, ab51502, Abcam), anti-Tbr2 (rabbit polyclonal, 1:500, ab23345, Abcam), anti-phospho-histone H3 (mouse monoclonal, 1:100, 05–598, millipore), anti-phospho-histone H3 (rabbit polyclonal, 1:100, 06–570, millipore), anti-γH2AX (mouse monoclonal, 1:100, 05–636, Millipore), anti-cleaved caspase-3 (rabbit polyclonal, 1:600, 9661S, Cell Signaling Technology, Danvers, MA, USA), anti-Tuj1 (mouse monoclonal, 1:100, 2G10, 05–559, upstate), anti-NeuN (rabbit polyclonal, 1:500, ab104225, Abcam), anti-DCX (rabbit polyclonal, 1:500, ab18723, Abcam), anti-Nestin (mouse monoclonal, 1:50, ab6142, Abcam), anti-Pax6 (rabbit polyclonal, 1:300, PRB-278P, Biolegend), anti-Sox2 (rabbit polyclonal, 1:100, AB5603, millipore), anti-Aspm (rabbit polyclonal, 1:500, del5 gift from Dr. Fujimoto [10]), and anti-γ-tubulin (mouse monoclonal, 1:1000, T6557, Sigma-Aldrich, Japan).

After three washes with PBS, the samples were incubated with Alexa488-, Alexa546- or Alexa647- conjugated anti-rabbit antibody (1:1000, Life Technologies, Japan), Alexa546- or Alexa647- conjugated anti-mouse antibody (1:1000, Life Technologies, Japan), or Alexa488 conjugated anti-rat antibody (1:1000, Life Technologies, Japan) for 1 h at room temperature. After three washes with PBS, the samples were incubated with DAPI solution (1.0 μg/mL in PBS) or YOYO1 solution (0.5 μg/mL in PBS) for 10 min at room temperature. The samples were then embedded with SlowFade (ThermoFisher, Japan).

The TUNEL reaction was carried out using the terminal deoxynucleotidyl transferase (TdT) (in situ cell death detection kit, TMR red, Roche, USA) according to the manufacturer's instructions. For the double-staining (TUNEL with neural markers) procedures, TUNEL staining was followed by immunostaining with the antibody against anti-NeuN, Tuj1, DCX, Tbr2 or Pax6. In order to stain BrdU incorporation, sections were rinsed with PBS to rehydration for 10 min at room temperature and incubated in 2 N HCl for 30 min at 37˚C to denature

the DNA, then washed with PBST (0.1% Tween/PBS) for 10 min 3 times. The sections were blocked for 2 hours at room temperature with a solution of 10% normal goat serum and 0.3% Tween in PBS. The sections were stained with anti-BrdU antibody (mouse monoclonal, 1:200, Bu-33 B2531, Sigma-Aldrich, Japan). In order to detect EdU incorporation, a Click-iT EdU cell proliferation kit (ThermoFisher scientific, USA) was used. All of the images acquired were obtained with a confocal laser microscope (Olympus FV1000).

For Aspm and γTub immunostaining, Z-stacks (1 μm steps × 3, 8-bit, 1024 ×1024 resolution) were observed through a Plan-Apochromat 63×/1.40 Oil DIC M27 using a ZEISS LSM 900 inverted confocal microscope. The brightness and contrast of all of the images were adjusted, and the maximum intensity projections of the z-stack images were taken using Fiji/ImageJ, version 1.53b.

## Image analysis

The images acquired with the Olympus FluoView FV1000 confocal microscope were processed with ImageJ/Fiji to adjust color and contrast. In order to conduct a quantitative analysis of the immunoreactive cells, at least three independent brain samples were analyzed at each embryonic stage (at E12.5, E14.5 and E16.5). The thickness of the cerebral cortex at E12.5, E14.5 and E16.5 and/or layer marker positive regions, which were clearly detectable with Satb2, Ctip2 and Tbr1 positive cells at E16.5 (Fig 1), was measured at three positions (left, center and right) within 250 μm-wide dorsal cortex regions, and the average thickness was calculated. In order to analyze the total number of immunoreactive cells, the cells were counted in each layer in each brain within the 250 μm-wide cortex regions, followed by calculation of the mean value obtained in three sections (Figs 2 and 3). Furthermore, measurements per genotype were calculated as the mean of at least three mice. In order to identify specific neural progenitors, apical progenitor cells (APs) were defined as the phospho-histone H3 (pHH3) positive cells located beneath the surface of the lateral ventricle, and basal progenitor cells (BPs) were defined as the pHH3 positive cells localized in the basal region of the ventricular zone (Fig 2) [17]. In order to analyze the distribution of the EdU positive cells in cerebral cortices at E16.5, the cortical layers were divided into 9 (WT) or 8 (cKO) identical bins of 100 pixels width using ImageJ (Fiji), and the number of cells in each bin was counted manually (WT: bin1-2 (ventricular zone (VZ)/subventricular zone (SVZ)), bin3-5 (intermediate zone (IZ)), bin6-9 (cortical plate (CP)); cKO: bin1-2 (VZ/SVZ), bin3-4 (IZ), and bin5-8 (CP)) (Fig 3). In order to measure the γH2AX positive cells, we used ImageJ plugin "Analyze Particles" and selected a signal size within 3.00–200.00 to remove the particles that were too small or big, and thus regarded as false positive signals. In order to analyze the distribution of TUNEL, γH2AX, CC3 positive cells and nuclear condensation, the cortical layers were divided into two (VZ, preplate (PP) at E12.5) or three layers (VZ, IZ and CP at E14.5, E16.5) and the total number was counted in each layer of the whole cortical regions, and the ratio (percentage) of the cells present in each layer was calculated. The density (cell number/area) was calculated to compare the cell density among the developmental brains.

For P7 brains (S1 Fig), a Nikon Ti Eclipse confocal microscope were used, and the images were processed with ImageJ/Fiji to adjust color and contrast. Whole brain images were obtained by tiling and stitching (6×6) the images captured with a 20× lens (2048×2048 pix). The high magnification images were cropped from the whole brain images.

## Statistical analysis

Each value was obtained from at least three independent samples. All data are shown as mean ± SEM. Prism 6 (Graph-Pad Software) was used for the analysis. For comparisons

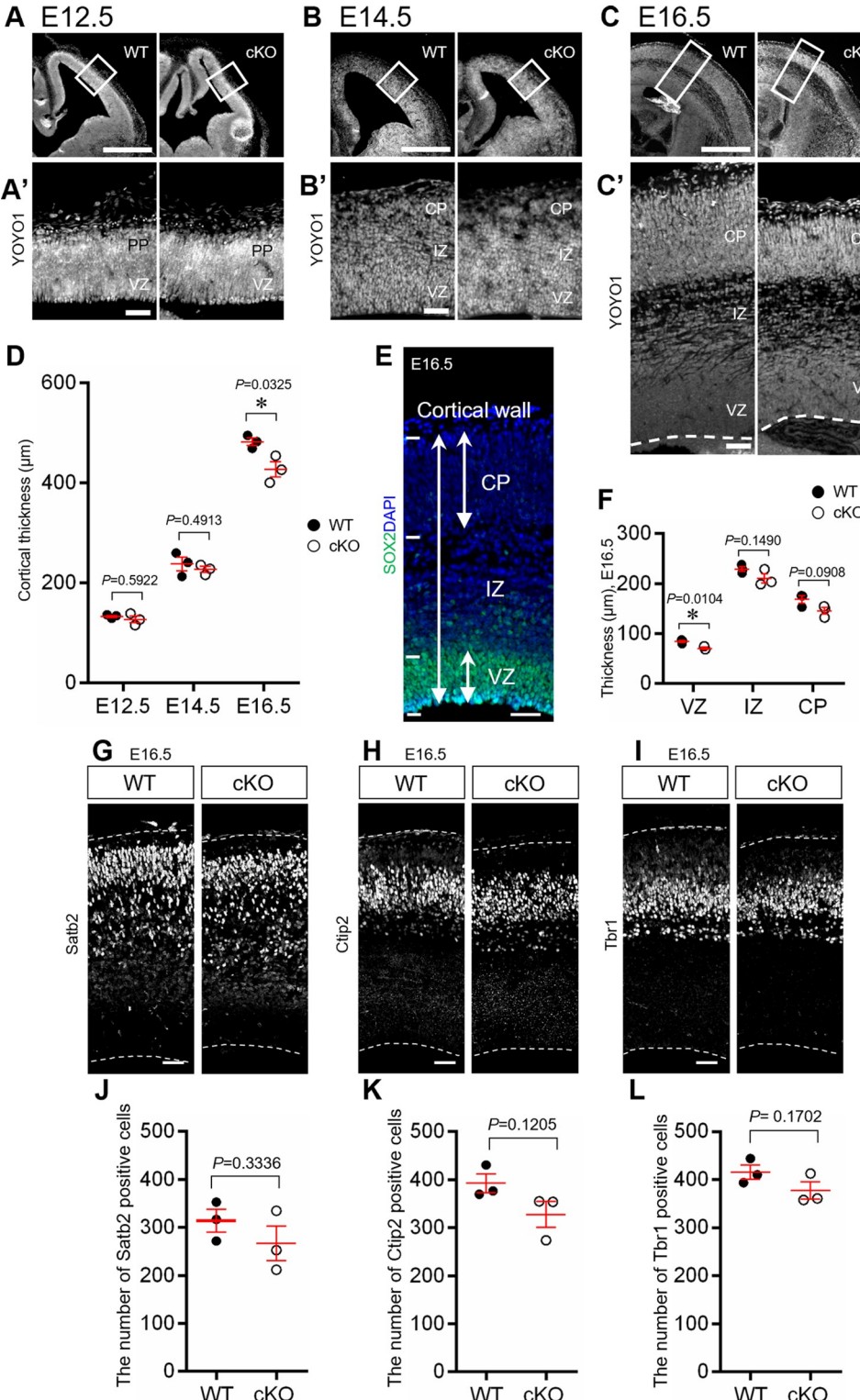

**Fig 1. Embryonic brain of Aspm cKO.** (A-C) Representative YOYO1 stained images of the embryonic cortices of the WT and Aspm cKO mice (NesCre;Aspm$^{\text{flox/flox}}$) at E12.5, E14.5 and E16.5. The scale bars represent 500 µm. (A'-C') Magnification of the boxed area in (A-C). The scale bars represent 50 µm. (D) Quantification of the thickness per unit area (A'-C') (250 µm radial columns) *: $P < 0.05$ (E12.5: WT 151.1 ± 2.0 µm, cKO 155.1 ± 6.5 µm, $P = 0.5922$, E14.5: WT 238.0 ± 13.5 µm, cKO 226.9 ± 5.8 µm, $P = 0.4913$, E16.5: WT 482.4 ± 7.5 µm, cKO 427.1 ± 15.5 µm, $P = 0.0325$,

Student's t-test, values represent mean ± SEM, n = 3 mice per genotype, ns: not significant) (E) SOX2 and DAPI stained image of E16.5 mouse cerebral cortex. The two head arrows indicate the CP, VZ and cortical wall. (F) Quantification of cortical wall and the CP region were defined based on YOYO1 staining and that of VZ was defined based on SOX2 staining. The IZ was defined as the value obtained by subtracting the CP and VZ from cortical wall. (VZ, WT 85.54 ± 2.2 μm, cKO 70.59 ± 2.1 μm, $P$ = 0.0104; IZ, WT 229.29 ± 4.9 μm, cKO 210.55 ± 9.3 μm, $P$ = 0.1490; CP, WT 159.51 ± 6.4 μm, cKO 140.03 ± 9.0 μm, $P$ = 0.0908, Student's t-test, values represent mean ± SEM, n = 3 mice per genotype) (G-I) Representative images of the WT and Aspm cKO Satb2, Ctip2 and Tbr1, respectively. Scale bar represents 50 μm. (J-L) Quantification of the cell number of the upper layer marker Satb2, the middle layer maker Ctip2 and the deep layer marker Tbr1, respectively, per unit area G-I (250μm radial columns). (J: WT 314.0 ± 23.4 cells, cKO 266.7 ± 36.2 cells, $P$ = 0.3336, K: WT 392.7 ± 19.3 cells, cKO 327.7 ± 26.8 cells, $P$ = 0.1205, L: WT 416.0 ± 14.7 cells, cKO 377.3 ± 17.9 cells, $P$ = 0.1702, Student's t-test, values represent mean ± SEM, n = 3 mice per genotype, ns: not significant).

between the WT and cKO, we employed the unpaired Student's t-test. Two-way ANOVA with Tukey's multiple comparisons test was conducted for comparison with the progenitor cell types, bins or cortical layers × genotype. The significance of the comparisons is represented on the graphs by asterisks; *$P$ < 0.05; **$P$ < 0.01; ***$P$ < 0.001 and ****$P$ < 0.0001, respectively.

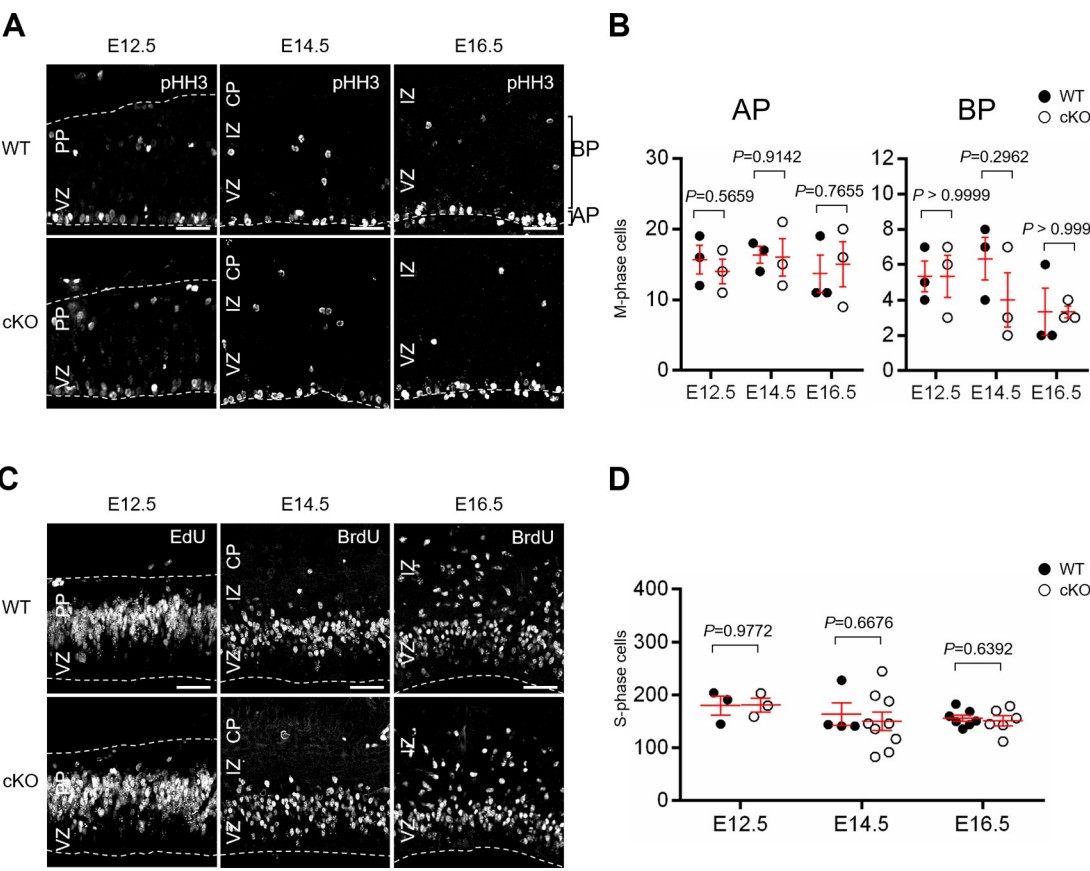

**Fig 2. Cell proliferation of NPCs during corticogenesis.** (A) Representative images of proliferative cells in the WT and Aspm cKO mice, which were positive for pHH3, in the developing cortex at E12.5, E14.5 and at E16.5. Scale bar: 50 μm. (B) The distribution of mitotic APs and BPs in the developing cortex of the WT and Aspm cKO mice. (AP, E12.5: $P$ = 0.5659, E14.5: $P$ = 0.9142, E16.5: $P$ = 0.7655; BP, E12.5: $P$ > 0.9999, E14.5: $P$ = 0.2962, E16.5: $P$ > 0.9999, Student's t-test, values represent mean ± SEM, n = 3 mice per genotype) (C) Representative images of proliferative cells in developing cortex from the WT and Aspm cKO mice at E12.5, E14.5 or at E16.5 one hour after the EdU (at E12.5) or BrdU (at E14.5 or E16.5) injections. Scale bar: 50 μm. (D) The number of EdU- or BrdU-positive cells in the embryonic cortex of the WT and Aspm cKO mice. (E12.5: $P$ = 0.9772 (WT: n = 3, cKO: n = 3), E14.5: $P$ = 0.6676 (WT: n = 4, cKO: n = 9), E16.5: $P$ = 0.6392 (WT: n = 7, cKO: n = 6), Student's t-test, values represent mean ± SEM).

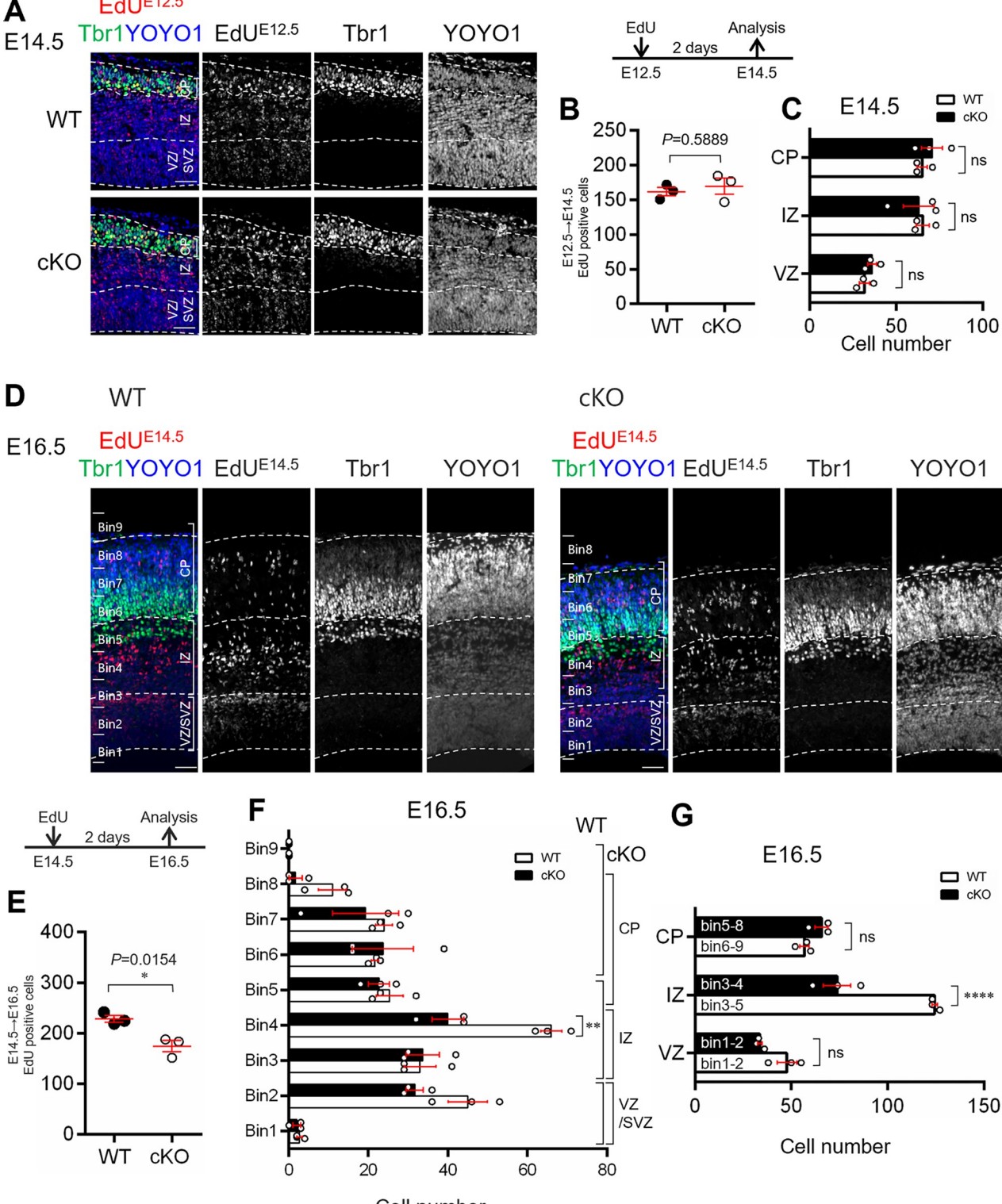

**Fig 3. Neurogenesis during murine corticogenesis.** (A, D) Co-staining images of Tbr1 (green), EdU (red) and YOYO1 (blue) in the WT and Aspm cKO mice at E14.5 (A) and E16.5 (D) cortices two days after the EdU injections (E12.5 and E14.5, respectively). Scale bar: 50 μm. (B, E) The number of EdU-positive cells per 250 μm. *$P < 0.05$ (B: WT 162.0 ± 6.1, cKO 169.7 ± 11.6, $P = 0.5889$, E: WT 228.7 ± 6.9, cKO 174.7 ± 11.4, $P = 0.0154$, Student's t-test, values represent mean ± SEM, n = 3 mice per genotype) (C) Comparison of the distribution of the EdU-positive cells among the cortical layers (cortical layers × genotype interaction $F_{(2, 12)} = 0.3488$ $P < 0.7125$; two-way ANOVA with Tukey's multiple comparisons test; ns: not

significant, values represent mean ± SEM, n = 3 mice per genotype). (F) Histogram of the EdU-positive cells, which were shown in C. (bins × genotype interaction F(8, 36) = 2.893 $P$ = 0.0135; two-way ANOVA with Tukey's multiple comparisons test; **$P$ < 0.01, values represent mean ± SEM, n = 3 mice per genotype). (G) Comparison of the distribution of the EdU-positive cells among the cortical layers (cortical layers × genotype interaction F(2, 12) = 27.82 $P$ < 0.0001; two-way ANOVA with Tukey's multiple comparisons test; ****$P$ < 0.0001, ns: not significant, values represent mean ± SEM, n = 3 mice per genotype).

## Results

### Cerebral cortical reduction in Aspm cKO was exhibited at the late embryonic stage

First, we generated brain specific Aspm knockout mouse (NesCre;$Aspm^{flox/flox}$, Aspm cKO) (S2 Fig) in order to examine whether the morphology of the embryonic cerebral cortex was affected by the loss of Aspm. Cortical thickness did not significantly change between the WT and cKO mice at E12.5 and E14.5 (E12.5, WT 151.1 ± 2.0 μm, cKO 155.1 ± 6.4 μm, $P$ = 0.5922; E14.5, WT 238.0 ± 13.5 μm, cKO 227.0 ± 5.8 μm, $P$ = 0.4913). However, a morphological difference was observed at E16.5, with a decrease in cortical thickness (WT 482.4 ± 7.5 μm, cKO 427.1 ± 15.5 μm, $P$ = 0.0325) (Fig 1A–1D).

We further analyzed how the changes were related to the layer construction of the cerebral cortex at E16.5. First, there was a significant decrease in the thickness of the VZ in Aspm cKO (Figs 1C', 1E and 1F and S3). Although the thickness of IZ or CP showed a slight decrease, the significance did not reach the level of significance as compared to WT (Figs 1C', 1E and 1F and S4).

The CP is composed of six layers, and each layer consists of neurons with different birthdates. Therefore, we examined the CP in more detail using specific markers for each cortical layer. In the mouse brains lacking Aspm, the number of cells immunoreactive for the upper layer marker (Satb2-positive, II/III and IV layer), the middle layer marker (Ctip2-positive, layer V) and for the deep layer marker (Tbr1-positive, layer VI) showed a slight decrease, but no significant differences, when compared to those of the WT mice, (Fig 1G–1L), suggesting that a normal cytoarchitecture of the cerebral cortex was formed even with a loss of Aspm.

These findings suggest that Aspm is important for NPC maintenance, not for neurogenesis or differentiation from NPCs to neurons at all embryonic stages.

### Loss of Aspm did not affect the proliferation of NPCs

Next, we conducted assays to determine whether or not the thinning of total cortical thickness and VZ thickness caused by Aspm deficiency is related to changes in the neural progenitor cell proliferation. In order to identify specific neural progenitors, apical progenitor cells (APs) were defined as the phospho-histone H3 (pHH3) positive cells located beneath the surface of the lateral ventricle, and basal progenitor cells (BPs) were defined as the pHH3 positive cells localized in the basal region of the ventricular zone (Fig 2) [17]. Immunofluorescence studies for pHH3, which is the mitotic marker, revealed no significant differences in the number of APs and BPs positive for pHH3 between the WT and Aspm cKO mice (Fig 2A and 2B).

Furthermore, similar to the pHH3 results, we examined the abundance of mitotic cells by thymidine analog labeling assay (EdU or BrdU). Mitotic cells in the S phase of the cell cycle were observed 1 hr after an intraperitoneal administration of EdU or BrdU in the basal VZ (Fig 2C and 2D). In both cases, there were no obvious differences in the mitosis of the NPCs between the WT and cKO mice at the early corticogenesis (at E12.5), the middle (at E14.5) or the late corticogenesis stages (at E16.5).

These results suggest that loss of Aspm does not affect the proliferation of the NPCs, but might affect NPCs pool during mouse neurogenesis.

## Loss of Aspm causes reduction in mid to late born neurons

In order to determine whether the loss of Aspm affects the generation of early- or late-born neurons, we performed an EdU-birthdating assay. Pregnant mice were given a single dose of EdU at E12.5 or E14.5, and two days later (i.e., E14.5 or E16.5, respectively), the abundance of EdU-positive cells was measured. The E12.5-labeled cells showed no significant differences between the WT and cKO mice (Fig 3A and 3B). The number of EdU-labeled cells with Tbr1-immunoreactivity in the CP, as well as the number of EdU-labeled cells in the IZ and VZ showed no significant difference between the WT and Aspm cKO brains at E12.5 (Fig 3C). In contrast, the number of cells labeled at E14.5 was significantly reduced, with most of them located in the VZ/SVZ or IZ areas in the cKO mice (Fig 3D and 3E), arguing that the main cell population affected by the loss of ASPM were NPCs rather than neurons. There was a significant reduction in the number of EdU-labeled cells distributed in the IZ, but not in the VZ and CP (Fig 3F and 3G). Although the most of the EdU-labeled cells in the CP showed immunoreactivity for Satb2, late born neurons, there were no significant differences in ratio of Tbr1$^+$/EdU$^+$, Ctip2$^+$/EdU$^+$, and Satb2$^+$/EdU$^+$ cells in the cerebral wall between the WT and Aspm cKO brains (S5 Fig).

These findings suggested that the loss of Aspm induced a reduction in the number of mid to late born neurons without any aberrant expression of cerebral layer-specific transcription factors in the developing cortex, which might be due to affected NPCs pool.

## Loss of Aspm led to an increase in apoptosis of NPCs

We conducted assays to determine whether the occurrence of apoptosis was related to the cortical thinning observed in association with the loss of Aspm (Fig 4). In the TUNEL assay for measuring apoptosis, TUNEL-positive cells were distributed in the VZ and PP at E12.5 (Fig 4A, 4B and 4J) and in the VZ, IZ and CP at E14.5 and E16.5 (Fig 4C–4F and 4J). Although the TUNEL-positive cells were significantly increased in the developing cortex in the cKO mice throughout corticogenesis (Fig 4G), postnatal brains showed no differences from WT brains (S1 Fig). The numerical density of the TUNEL-positive cells in the developing cortex (TUNEL index) decreased as the embryonic stage proceeded (Fig 4H and 4I). Furthermore, the TUNEL-index was significantly higher in the VZ at E12.5 and 14.5, but not at E16.5 in the cKO mice as compared to WT mice (Fig 4J). Thus, we found an increased number of apoptosis instances in the developing cortex obtained from Aspm cKO mice and the apoptosis was higher in the VZ at the early to the middle phase of the corticogenesis, compared with the later corticogenesis.

In order to confirm the features of the apoptosis, we employed nuclear morphology, including nuclear condensation and DNA fragmentation, and the molecular signatures were evaluated by immunohistochemistry for CC3 (S6 Fig). The CC3-positive cells and/or nuclear condensation were significantly increased in the cKO mice at E12.5, distributed in the VZ and PP. The CC3-positive cells and/or nuclear condensation were significantly increased at E14.5 and E16.5 as well. Furthermore, a significant increase in the numerical density of CC3-positive cells (CC3 index) was revealed in the VZ at E14.5 and in the IZ at E16.5 in samples obtained from the developing cortex of the cKO mice. The numerical density of nuclear condensation (nuclear index) showed a significant increase in the VZ at E14.5 in cKO mice.

Next, we conducted tests to determine whether the apoptosis was induced in neural progenitor cells, intermediate progenitors, or neurons by the loss of Aspm (S7 Fig). TUNEL-positive cells expressed the neuronal markers NeuN, Tuj1, and DCX (S7A–S7C and S7A'–S7C' Fig), but not the APs marker Pax6 or BPs marker Tbr2 (S7D–S7E and S7D'–S7E' Fig). In

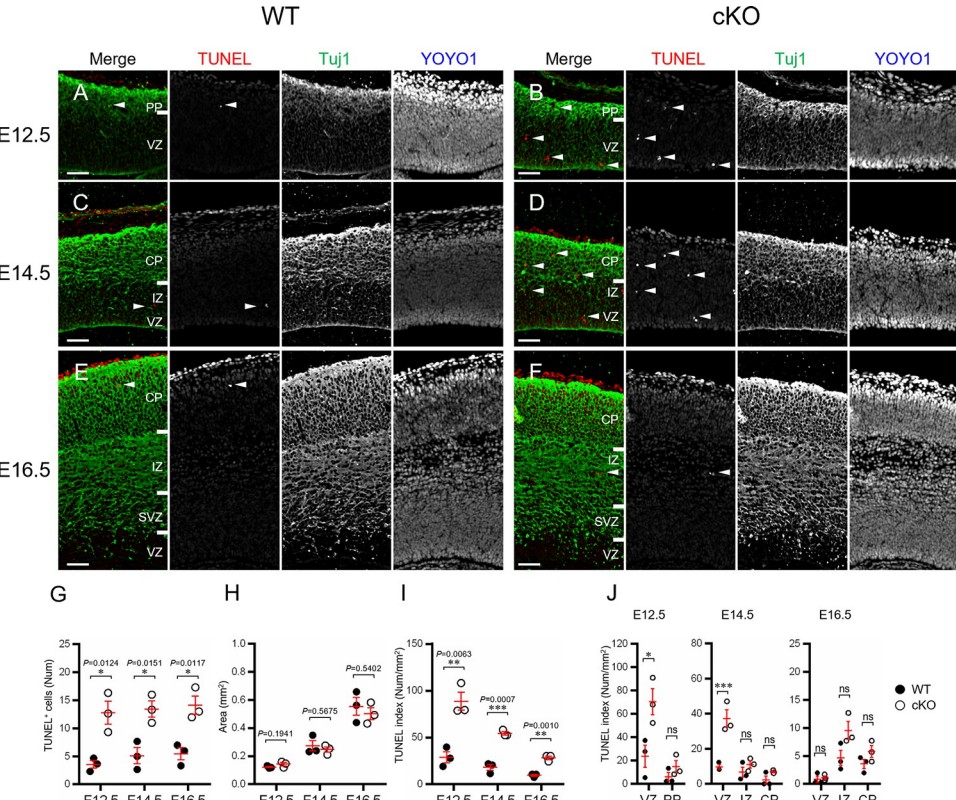

**Fig 4. TUNEL assay.** (A-F) Representative double staining images (TUNEL and the neuronal marker Tuj1) of the WT and Aspm cKO embryonic cortices at E12.5 (A, B), E14.5 (C, D) and E16.5 (E, F). Scale bars represent 50 μm. (G-I) TUNEL index (I: *Num/Area*) is the number of TUNEL-positive cells (G: *Num*) in a cerebral cortex divided by the unit area (H: *Area*). (G: E12.5 $P = 0.0124$, E14.5 $P = 0.0151$, E16.5 $P = 0.0117$; H: E12.5 $P = 0.1941$, E14.5 $P = 0.5675$, E16.5 $P = 0.5402$; I: E12.5 $P = 0.0063$, E14.5 $P = 0.0007$, E16.5 $P = 0.0010$, Student's t-test, $^*P < 0.05$; $^{**}P < 0.01$ and $^{***}P < 0.001$, values represent mean ± SEM, n = 3 mice per genotype) (J) Distribution of the TUNEL index in the developing cortex divided into the VZ, IZ, PP and/or CP. (E12.5: cortical layers × genotype interaction $F_{(1, 8)} = 5.709$ $P = 0.0439$, E14.5: cortical layers × genotype interaction $F_{(2, 12)} = 11.96$ $P = 0.0014$, E16.5: cortical layers × genotype interaction $F_{(2, 12)} = 2.471$ $P = 0.1263$; two-way ANOVA with Tukey's multiple comparisons test; $^*P < 0.05$; $^{**}P < 0.01$; $^{***}P < 0.001$ and $^{****}P < 0.0001$, ns: not significant, n = 3 mice per genotype).

CC3-positive cells, the expression was consistent with the neural progenitor cell marker Nestin (S7F and S7F' Fig).

Judging from these results, we conclude that Aspm expression is required for the maintenance of neural progenitor cells and neurons and plays a critical role in during neurogenesis.

## DNA-damaged cells are increased along with the loss of Aspm in developing cortex

Next, we conducted assays to determine whether DNA damage has relevance to increased apoptosis in developing cortex along with the loss of Aspm. Immunofluorescence for the DNA damage marker, γH2AX, revealed that the number of DNA-damaged cells significantly increased at E12.5 and E14.5, that is, in the early- and middle-phase of corticogenesis in cKO mice (Fig 5).

The numerical density of γH2AX-immunoreactive cells (γH2AX index) tended to increase in the VZ, PP, IZ and in the CP from the early to the late phases of corticogenesis in the cKO mice brains, although the difference did not reach the level of significance. This tendency is

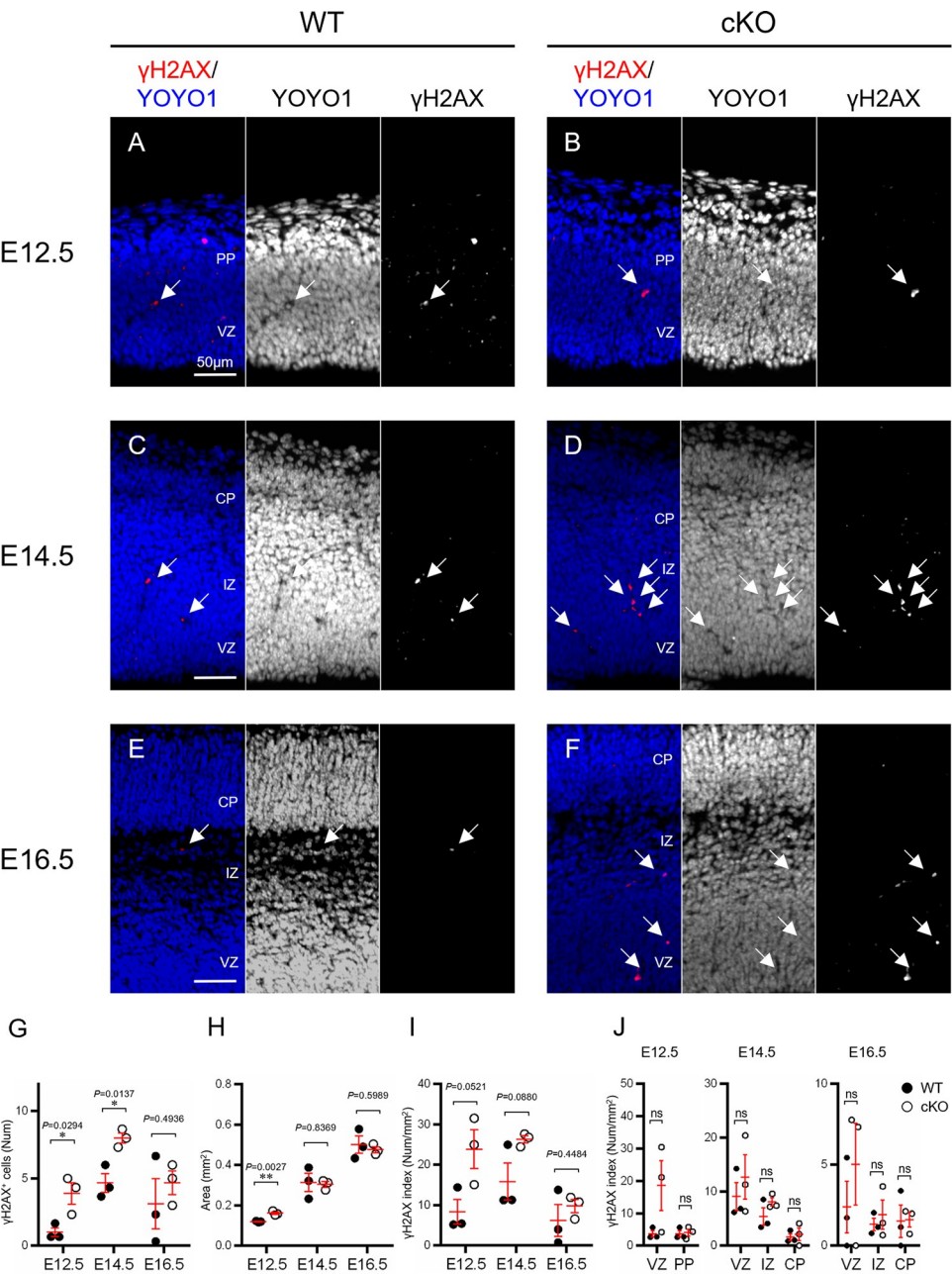

**Fig 5. Immunohistochemistry for γH2AX.** (A-F) Representative immunofluorescence images (γH2AX (red) and YOYO1 (blue) of embryonic cortex in the WT and Aspm cKO mice at E12.5 (A, B), E14.5 (C, D) and E16.5 (E, F). The number of γH2AX-positive cells were counted in the developing cortex. Arrows indicate γH2AX-positive cells (red). Scale bars represent 500 μm. (G) The number of γH2AX positive cells in cerebral cortex (E12.5: WT1.00 ± 0.33 cells, cKO 3.89 ± 0.80 cells $P$ = 0.0294, E14.5 WT 4.67 ± 0.69 cells, cKO 8.00 ± 0.39 cells $P$ = 0.0137, E16.5: WT 3.11 ± 1.87 cells, cKO 4.67 ± 0.88 cells $P$ = 0.4936, Student's t-test, *$P$ < 0.05, values represent mean ± SEM, n = 3 mice per genotype) (H) The area of cerebral certex which cell number was counted. (E12.5: WT 0.12 ± 0.0 mm$^2$, cKO 0.16 ± 0.01 mm$^2$, $P$ = 0.0027, E14.5: WT 0.31 ± 0.04 mm$^2$, cKO 0.30 ± 0.01 mm$^2$, $P$ = 0.8369, E16.5: WT 0.50 ± 0.04 mm$^2$, cKO 0.48 ± 0.01 mm$^2$, $P$ = 0.5989, Student's t-test, **$P$ < 0.01, values represent mean ± SEM, n = 3 mice per genotype) (I) γH2AX index (Num/Area) is the number of γH2AX-positive cells (G) in a cerebral cortex divided by the unit area (H). (E12.5: WT 8.27 ± 3.04 cells/mm$^2$, cKO 23.89 ± 4.79, $P$ = 0.0521, E14.5: WT 15.87 ± 4.56 cells/mm$^2$, cKO 26.32 ± 0.94, $P$ = 0.0880, E16.5: WT 6.21 ± 3.93 cells/mm$^2$, cKO 9.78 ± 1.62, $P$ = 0.4484, ns: not significant, values represent mean ± SEM, n = 3 mice per genotype) (J) Distribution of γH2AX index in the developing cortex divided into the VZ, IZ, PP and/or CP. (E12.5: cortical layers × genotype interaction $F(1, 8)$ = 3.387 $P$ = 0.1030, E14.5: cortical layers × genotype interaction $F(2, 12)$ = 0.2642 $P$ = 0.7722, E16.5: cortical layers × genotype interaction $F(2, 12)$ = 0.4965 $P$ = 0.6206; two-way ANOVA with Tukey's multiple comparisons test; *$P$ < 0.05, ns: not significant, n = 3 mice per genotype).

consistent with previous results regarding apoptosis. Furthermore, the double immunofluorescence of γH2AX with the neural progenitor cell marker, Pax6, revealed that the cells with DNA damage were NPCs expressing Pax6 (S8 Fig). However, Pax6 expression was not observed in cells with high nuclear condensation. This result suggests that the Pax6 protein was not detected in NPCs with strong DNA damage due to the loss of Aspm. In addition, the thickness of VZ and the number of NPCs, immunoreactive for PAX6, showed a decrease at E16.5 in cKO compared to WT (S3 Fig).

Altogether, these results suggest that the loss of Aspm causes an increase in the number of cells with DNA damage in developing murine cortex, resulting in a reduction of VZ at the later corticogenesis stage.

## Discussion

In this study, we compared brain-specific Aspm-knockdown mice with normal control mice and the results provided important insights into how the loss of Aspm causes abnormal brain formation in developing cortex. These findings suggest that the microcephaly associated with MCPH5 might be aggravated by increased apoptosis in NPCs, but not by aberrant proliferation of NPCs during cortical development, although we can't exclude that there are other ways of aberrant division such as altered ratio of self-renewal vs. neurogenic divisions.

### Loss of Aspm reduced cerebral cortical thickness with a thinning of all cortical layers

The developing cerebral wall exhibited a significant reduction in thickness at E16.5 in the cKO mice, which was associated with a significant thinning of the ventricular zone and a reduction in cell numbers in all of the cortical layers (Fig 1). These findings suggest that the thinning of the cortex might be due to poor neurogenesis at mid to late embryonic stages and the reduction in VZ-thickness at E16.5 might impair further gliogenesis (Figs 1F and S3). Reduced brain size is thought to be due to a reduced number of total neurons caused by decreased proliferation of neural progenitors during fetal development [3,4]. Fish et al. [13] reported that Aspm-downregulation, using RNA interference, induced the alteration in the cleavage plane orientation in neuroepithelial cells of embryonic telencephalon, followed by a reduction in the number of neuroepithelial progenitor cells and microcephaly. In the present study, we did not find any aberrant cleavage plane orientation in the neuroepithelial cells and the mitosis of the NPCs was not affected in the cortex of the cKO mice when we focused on the S and M phase of the cell cycle (Fig 2). Therefore, it is highly likely that cell loss might occur not only in NPCs, but also in postmitotic neurons, i.e., neuroblasts in the VZ/SVZ, migrating neurons in the IZ or immature neurons after reaching the CP. These observations suggest that this cell loss could play some role in building the normal cortical architecture, except for uniform thinning, as observed in the cKO mice. It is worth noting that the EdU-birthdating assay revealed that the number of cells which were born at E14.5 was significantly reduced, with a significant reduction in IZ from the cKO mice (Fig 3), which indicated that the loss of Aspm might reduce the number of mid to later born neurons in IZ. It has been reported that the deletion of Aspm induced prematurely exit of the cell cycle and early differentiation of the cerebellar granule neuron progenitors, resulting in a decrease of terminally differentiated granule neurons in the internal granule layer [18]. Although the present analyses were conducted in regard to the cerebral cortical development related to the loss of Aspm, the reduced mid to later born neurons might have been induced by impaired mitotic progression due to an earlier exit of the cell cycle with the accelerated differentiation of NPCs.

## Loss of Aspm is involved in apoptosis induction during embryonic neocortex formation, not in postnatal brain

The knockout of Aspm in NPCs increased apoptosis in all of the stages of cortical development compared to normal brains (Fig 4), but postnatal brains showed no differences from normal brains (S1 Fig). Considering reports [12,19,20] that Aspm contributes to the stabilization of symmetric divisions in mitotic NPCs, it is likely that the expression of Aspm in NPCs is indispensable for maintaining the degree of stemness which enables neuron production. We found that the frequency of apoptosis was exclusively higher in the VZ in the early to the mid corticogenesis stages, resulting in a reduction of the neural progenitor cell pool required for neurogenesis (Figs 1F and S3).

Aspm-deficient cerebellar granule neuron progenitors show impaired mitotic progression, altered patterns of division orientation and differentiation, and increased DNA damage, which causes progenitor attrition through apoptosis [18]. Similar mechanisms might be involved in the abnormal corticogenesis in our Aspm cKO.

Jayaraman et al. [21] reported that loss of Wdr62, Aspm or both impairs centriole duplication, correlating with the severity of microcephaly, and leads to a loss of centrosomes and cilia in the developing brain, and accordingly, microcephaly has been described as a 'centriolopathy'. They showed an expansion of basal progenitors beyond the germinal zones at mid-neurogenesis, suggesting regulation of cell fate leading to precocious formation of basal progenitors rather than merely mitotic progression in Emx1-Cre; Aspm-flox. In addition, they showed little-to-no apoptosis in the Aspm-/- mice brains. We did not find any remarkable delamination of basal progenitor cells beyond the germinal zones at E14.5, using birthdate analyses (Fig 3). Furthermore, the number of apoptotic cells was significantly higher in all stages of corticogenesis in our Aspm cKO mice brains (Figs 4, 5 and S6), suggesting that the microcephaly might be followed by cell loss via apoptosis rather than proliferation and/or cell fate dysregulation. The reasons for these different results are unknown, but they might have been due to differences in the Aspm KO mice and experiment methods employed, including the ages or markers analyzed.

Johnson MB et al. [22] published an excellent study showing that Aspm KO ferrets, a gyrencephalic animal, exhibited severe microcephaly, reflecting reduced cortical surface area without any significant change in cortical thickness, as in human patients. The mutant ferret fetal cortex displayed a massive premature displacement of ventricular radial glial cells (VRG), the most undifferentiated cell type, to the outer subventricular zone (OSVZ), and a more differentiated progenitor, resulting in a loss of "cortical units". This finding is similar to that in another previous report [21]. Unfortunately, we did not find remarkable displaced progenitors, or severe microcephaly in the present Aspm cKO brains, which is most likely because of that mice, agyrencephalic animal, have few OSVG, Tbr2-immunoreactive BPs.

## Aspm is required for stabilization of symmetric NPC division

It has been reported that Aspm expression and BRCA1 expression are related. For example, in experiments using cultured cancer cells, when Aspm expression was suppressed, BRCA1 expression was also decreased and the reverse has also been confirmed [23–25]. Furthermore, both BRCA1 and BRCA2, DNA damage repair factors, are important for the homologous recombination (HR)-mediated DNA damage repair pathway [26,27]. Recently, HR has been reported to be a DNA damage repair pathway that repairs DNA damage caused by symmetric division of NPC [27]. Symmetric division of NPC is a mode of division seen early in the cortical formation process [20]. Based on these observations, we propose the hypothesis that the DNA damage that occurs in the early phase of brain formation is repaired via HR by DNA damage repair molecules such as BRCA1 and BRCA2.

Since our data showed that the loss of Aspm expression results in a greater number of apoptosis instances in NPC during early cortical formation (Figs 4 and S6), it is worth speculating how Aspm can stabilize symmetric divisions in NPC, through centrosome biogenesis and/or spindle organization and positioning during mitosis. Novorol et al. reported that there was increased apoptosis in MCPH models, including Aspm knockdown zebrafish [28], which appeared to be secondary to the mitotic defects, such as abnormal prometaphase progression. In the current study, we only examined the proliferation of NPCs only during the S and M phases, but we found no abnormalities despite the loss of Aspm, and thus, it is likely that DNA-damage may induce early cell death after the metaphase or postmitotic neurons.

## Conclusion

The results of this study demonstrated that the loss of Aspm is involved in increasing the number of apoptosis instances during cerebral cortical development. The increase in apoptosis occurred in both NPCs and neurons, particularly from the early stages of cortical formation, and gradually decreased in the late stages. These findings suggest that Aspm deficiency can induce apoptotic loss of NPCsand immature neurons, leading to decreased NSCs pool and neuronal loss and ultimately to phenotypic change (i.e., cortical thinning) as found in MCPH.

## Supporting information

**S1 Fig. TUNEL staining of postnatal mouse brain at P7.** (A-B) representative TUNEL staining images of WT and Aspm cKO postnatal brains at P7. Scale bars represent 1 mm. (A'-B') Higher magnification of the boxed area in A and B, respectively. Arrowheads indicated TUNEL-positive cells.
(TIF)

**S2 Fig. Aspm cKO and wild type mice.** This figure shows the Aspm cKO and wild type mice used in this study (see Materials and Methods). (A) The mouse Aspm gene targeting construct (B) Genotyping of the conditional Aspm knockout mouse (C) Immunostaining for Aspm (red), γTub (centrosome; green) and DNA (blue) in the NPCs of the wild type and conditional knockout (NesCre;Aspm$^{f/f}$) E14.5 mice brains.
(TIF)

**S3 Fig. Thickness of the ventricular zone in developing mouse cortex.** (A-C) Representative SOX2 staining images (green) of embryonic cortex in the WT and Aspm cKO mice at E12.5 (A), E14.5 (B) and E16.5 (C). (D, E) Quantification of the thickness of the ventricular zone and the cell number immunoreactive for SOX2 at E12.5, E14.5 and E16.5. Arrows indicate the thickness measurements. *: $P < 0.05$, ns: not significant (D: E12.5: $P = 0.6180$, E14.5: $P = 0.8887$, E16.5: $P = 0.0104$, D: E12.5: $P = 0.2012$, E14.5: $P = 0.4038$, E16.5: $P = 0.0842$, Student's t-test, values represent mean ± SEM, n = 3 mice per genotype).
(TIF)

**S4 Fig. Cortical layer thickness in developing mouse cortex.** (A-C) Representative immunofluorescence images (Tbr1 (green) and YOYO1 (blue)) of embryonic cortex in the WT and Aspm cKO mice at E12.5 (A), E14.5 (B) and E16.5 (C). (D, E) Quantification of the thickness of the cortical wall and cortical layer (PP at E12.5 or CP at E14.5 and E16.5). Arrows indicate the thickness measurements. *: $P < 0.05$, ns: not significant (D: E12.5: $P = 0.5922$, E14.5: $P = 0.4913$, E16.5: $P = 0.0325$, E: E12.5: $P = 0.2291$, E14.5: $P = 0.1897$, E16.5: $P = 0.0906$, Student's t-test, values represent mean ± SEM, n = 3 mice per genotype).
(TIF)

**S5 Fig. Identification of EdU labeled cells by neural marker.** (A-C) Representative images of EdU birthdating assay in WT E16.5 cortices two days after EdU injections (E14.5). Scale bar: 50 μm. (D-F) Higher magnification of the boxed area in A-C. Scale bar: 50 μm. Yellow-filled arrowheads indicated Tbr1$^+$/EdU$^+$, Ctip2$^+$/EdU$^+$ or Satb2$^+$/EdU$^+$ double positive cells, respectively and white-open arrowheads indicated EdU single positive cells. (G-I) Quantitative assessment of cell type of EdU positive cells at E16.5 two days after EdU labeling at E14.5. (G: WT n = 492 cells, 8.7 ± 2.0%, cKO n = 408 cells, 9.9 ± 3.5%, *P* = 0.7606, H: WT n = 592 cells, 3.4 ± 1.8%, cKO n = 451 cells, 3.4 ±1.8%, *P* = 0.3744, I: WT n = 458 cells, 76.6 ± 12.8%, cKO n = 460 cells, 86.9 ± 6.0%, *P* = 0.5079. Student's t-test, values represent mean ± SEM, n = 3 per genotype).
(TIF)

**S6 Fig. Immunohistochemistry for cleaved caspase 3 and nuclear condensation.** (A-F) Representative immunohistochemistry images (cleaved caspase 3 (CC3) and nuclear condensation of the embryonic cortex at E12.5 (A, B), E14.5 (C, D) and E16.5 (E, F) in the WT and Aspm cKO mice. The number of CC3-positive cells and/or nuclear condensation were counted in developing cortex, separated into the ventricular zone (VZ), the preplate (PP), the intermediate zone (IZ), and the cortical pate (CP). The scale bars represent 100 μm. (A'-F') Higher magnification of the boxed area in (A-F). Arrows indicated condensed nuclei, which were shown as YOYO1 higher intense singals. The scale bars represent 50 μm. (G, H) CC3 index (H: *Num/Area*) is the number of CC3-positive cells (G: *Num*) in a cerebral cortex section divided by the unit area (*Area*). (G: E12.5 *P* = 0.0475, E14.5 *P* = 0.0075, E16.5 *P* = 0.0015; H: E12.5 *P* = 0.0055, E14.5 *P* = 0.0076, E16.5 *P* = 0.0045, Student's t-test, *$*P$ < 0.05; **$*P$ < 0.01, values represent mean ± SEM, n = 3 mice per genotype). (I) Distribution of the CC3 index in the developing cortex divided into the VZ, IZ, PP and/or CP (E12.5: cortical layers × genotype interaction $F(1, 8) = 0.8502$ *P* = 0.3835, E14.5: cortical layers × genotype interaction $F(2, 12) = 1.437$ *P* = 0.2757, E16.5: cortical layers × genotype interaction $F(2, 12) = 2.748$ *P* = 0.1041; two-way ANOVA with Tukey's multiple comparisons test; *$*P$ < 0.05; **$*P$ < 0.01, ns: not significant, n = 3 mice per genotype) (J, K) Nuclear index (K: *Num/Area*) is the number of nuclear condensation instances (J: *Num*) in a cerebral cortex section divided by the unit area (*Area*). (J: E12.5 P = 0.0380 E14.5 *P* = 0.0405, E16.5 *P* = 0.0036; K: E12.5 *P* = 0.0051, E14.5 *P* = 0.0693, E16.5 *P* = 0.0016, Student's t-test, *$*P$ < 0.05; **$*P$ < 0.01, values represent mean ± SEM, n = 3 mice per genotype). (L) Distribution of Nuclear index in the developing cortex divided into the VZ, IZ, PP and/or CP (E12.5: cortical layers × genotype interaction $F(1, 8) = 2.250$ *P* = 0.1720, E14.5: cortical layers × genotype interaction $F(2, 12) = 16.77$ *P* = 0.0003, E16.5: cortical layers × genotype interaction $F(2, 12) = 2.384$ *P* = 0.1343; two-way ANOVA with Tukey's multiple comparisons test; *$*P$ < 0.05; **$*P$ < 0.01; ***$*P$ < 0.001 and ****$*P$ < 0.0001, ns: not significant, n = 3 mice per genotype).
(TIF)

**S7 Fig. TUNEL staining and neuronal makers of the Aspm cKO mice.** Representative double immunofluorescence images of TUNEL and cell type-specific markers (NeuN (A, A'), TuJ1 (B, B'), DCX (C, C'), Tbr2 (D, D') or Pax6 (E, E')) in the embryonic cortex of Aspm cKO mice at E14.5. Scale bar: 100 μm (see Materials and Methods). Most of the TUNEL-positive cells were immunoreactive for neuronal markers NeuN (A') and Tuj1 (B') (filled arrowhead) and immature neuronal marker DCX (C'). In contrast, no double labeling was detected for the TUNEL or the intermediate progenitor cell marker Tbr2 (D') or neural progenitor cell marker Pax6 (E') (open arrowheads). Representative double immunofluorescence images of cleaved caspase 3 (CC3) and neural progenitor cell marker (Nestin (F, F')) in the embryonic cortex of Aspm cKO mice at E14.5, in which the filled arrowhead indicated Nestin/CC3 double positive

cells. Scale bar: 10 μm.
(TIF)

**S8 Fig. DNA damages occurred in NPCs of Aspm cKO.** (A) Representative double immuno-fluorescence images of the DNA damage marker (γH2AX) and the neural progenitor cell marker (Pax6) in the murine embryonic cortex of Aspm cKO mice at E14.5. (B-E) Higher magnification of the boxed area in (A). In γH2AX-positive cells, Pax6 was positive in NPCs with weak nuclear condensation (filled arrowheads), although Pax6 was negative in cells with a high nuclear condensation (open arrowheads). Scale bar: 100 μm (A), 10 μm (B-E).
(TIF)

**S1 Raw images. Raw images of genotyping result (S2 Fig).** This is an original full-size image of an agarose gel in which the genotyping PCR products from the transgenic individuals have been size fractionated (3% agarose/TAE, 100V, 20min). Each lane corresponds to one individual. PCR products to detect Aspm$^{flox}$ and Cre were loaded in the top and bottom lanes, respectively. f and + indicate floxed and wild-type alleles, respectively. In the bottom lanes, + and—indicate the presence or absence of the Cre cassette. S2B Fig shows the cropped image of a set of lanes labeled WT and cKO.
(PDF)

## Author Contributions

**Conceptualization:** Kyoko Itoh.

**Data curation:** Madoka Tonosaki, Takeshi Yaoi, Kyoko Itoh.

**Funding acquisition:** Madoka Tonosaki, Kyoko Itoh.

**Investigation:** Madoka Tonosaki, Takeshi Yaoi.

**Methodology:** Madoka Tonosaki, Akira Fujimori.

**Project administration:** Kyoko Itoh.

**Resources:** Akira Fujimori.

**Supervision:** Kyoko Itoh.

**Validation:** Kyoko Itoh.

**Writing – original draft:** Madoka Tonosaki.

**Writing – review & editing:** Kyoko Itoh.

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
