## [Decision Letter · Decision Letter 0]

16 Apr 2023

PONE-D-23-02973Loss of Aspm causes increased apoptosis of developing neuronal cells during mouse cerebral corticogenesisPLOS ONE

Dear Dr. Itoh,

Thank you for submitting your manuscript to PLOS ONE. After careful consideration, we feel that it has merit but does not fully meet PLOS ONE’s publication criteria as it currently stands. Therefore, we invite you to submit a revised version of the manuscript that addresses the points raised during the review process.

Both reviewers agree that there are several technical issues and problems with data interpretation that need to be addressed, before the article is suited for publication. Please provide a detailed response to reviewers comments in your next submission

We look forward to receiving your revised manuscript.

Kind regards,

Carlos Oliva, PhD

Academic Editor

PLOS ONE

Journal Requirements:

"This work was supported by JSPS KAKENHI Grant Numbers JP25293240 (Kyoko Itoh), JP16K19689 and JP18K15683 (Madoka Tonosaki)."

"This work was supported by JSPS KAKENHI Grant Numbers JP25293240 (Kyoko Itoh), JP16K19689 and JP18K15683 (Madoka Tonosaki)."

"The authors hereby declare no conflict of interest associated with this study."

7. PLOS ONE now requires that authors provide the original uncropped and unadjusted images underlying all blot or gel results reported in a submission’s figures or Supporting Information files. This policy and the journal’s other requirements for blot/gel reporting and figure preparation are described in detail at https://journals.plos.org/plosone/s/figures#loc-blot-and-gel-reporting-requirements and https://journals.plos.org/plosone/s/figures#loc-preparing-figures-from-image-files. When you submit your revised manuscript, please ensure that your figures adhere fully to these guidelines and provide the original underlying images for all blot or gel data reported in your submission. See the following link for instructions on providing the original image data: https://journals.plos.org/plosone/s/figures#loc-original-images-for-blots-and-gels. 

8. We note that you have included the phrase “data not shown” in your manuscript. Unfortunately, this does not meet our data sharing requirements. PLOS does not permit references to inaccessible data. We require that authors provide all relevant data within the paper, Supporting Information files, or in an acceptable, public repository. Please add a citation to support this phrase or upload the data that corresponds with these findings to a stable repository (such as Figshare or Dryad) and provide and URLs, DOIs, or accession numbers that may be used to access these data. Or, if the data are not a core part of the research being presented in your study, we ask that you remove the phrase that refers to these data.

Reviewers' comments:

Reviewer's Responses to Questions

**Comments to the Author**

1. Is the manuscript technically sound, and do the data support the conclusions?

Reviewer #1: Partly

Reviewer #2: Partly

2. Has the statistical analysis been performed appropriately and rigorously? 

Reviewer #1: Yes

Reviewer #2: Yes

3. Have the authors made all data underlying the findings in their manuscript fully available?

Reviewer #1: Yes

Reviewer #2: Yes

4. Is the manuscript presented in an intelligible fashion and written in standard English?

Reviewer #1: Yes

Reviewer #2: Yes

5. Review Comments to the Author

Reviewer #1: The manuscript by Tonosaki et al., entitled "Loss of Aspm causes increased apoptosis of developing neuronal cells during mouse cerebral corticogenesis" described the role of ASPM in embryonic murine brain cortical development, suggesting that the loss of ASPM reduced cortical thickness by increased apoptosis of neural precursor cells and postmitotic neurons. Although most of the experiments suggested this, there are some data interpretation inconsistencies. For example, the authors mentioned that the reduced cortical thickness is due to early neurogenesis since the Tbr1-associated layer was the only one reduced. However, they do not show changes in cortical thickness at early neurogenesis, only at late neurogenesis. Furthermore, as the Tbr1 neurons are the first to be generated, how can they see cortical defects only at E16.5 and not at E12.5 or E13.5? Or, if the changes are only seen at E16.5, why are there no changes in Satb2 neurons? Also, the authors did not cite or discuss important published articles such as Johnson et al., 2018. Nature or Williams et al., 2015. Development, where they mentioned that loss of Aspm is also associated with apoptosis and important for brain development.

Comments:

1. Fig. 1E does not explain how VZ/, IZ, or CP was defined.

2. In Fig. 1F-K it is not explained how the authors defined the areas measured. For example, for SATB2, several cells spread across the cortex are SATB2 positive. Then, what parameter is used by the authors to establish the right quantification area? Quantifying the number of SATB2, CTIP2 and TBR1 positive cells would be more appropriate. Also, the authors did not state the age of the samples analyzed. If the samples are embryonic, how can they define specific cortical layers since they have not been adequately established yet?

3. The authors mentioned, "Since the deep layer neurons are composed of early-born neurons, these findings suggest that the thinning of the cortex in late cortical formation might be due to poor neurogenesis at early embryonal stages". If only early neurogenesis is affected, how later is neurogenesis not? Depleting NPCs at early stages should also affect later neurogenesis as the NPC pool is reduced. Also, it is possible that the loss of ASPM only affects direct neurogenesis but not indirect neurogenesis. However, the authors did not test this possibility.

4. The authors mentioned that "Immunofluorescence studies for pHH3, which is the mitotic marker, revealed no significant differences in the abundance of apical progenitor cells (APs) and basal progenitor cells (BPs) between the WT and Aspm cKO mice (Figs 2A and 2B)". How did the authors determine apical or basal progenitors? The authors assume the nature of these progenitors based on their spatial location. However, it is highly recommended that these experiments be performed using proper cell markers such as Pax6 or Tbr2 and colocalization studies.

5. In Fig. 2C-D, it is unclear whether the absence of proliferative disruption is specific to a particular progenitor population since the authors did not use any cell-specific marker. Therefore, it is highly recommended that these experiments be performed using proper cell markers such as Pax6 or Tbr2 since the authors state that only early neurogenesis is affected.

6. The authors mentioned, "Pregnant mice were given a single dose of EdU at E12.5 or E14.5, and two days later (i.e., E14.5 or E16.5, respectively), the abundance of EdU-positive neurons was measured. Most of the cells, EdU-labeled at E12.5, were observed in the superficial area corresponding to the CP, and EdU-positive cells were found in postmitotic neurons showing Tbr1-immunoreactivity". However, TBR1 immunostaining was not performed. So then, how can the authors ensure that EdU staining corresponds to postmitotic neurons? Also, the authors made similar conclusions on Fig. 3C-F but did not perform immunostainings for specific neuronal populations such as Satb2, Ctip2 or Tbr1.

7. The authors do not have data to support the following statement: "These findings suggest that the loss of Aspm induced a reduction in the number of mid-born neurons under radial migration" since they did not perform neuronal migration experiments.

8. Fig. 4A-H are not cited in the text.

9. In Figure 5, it is unclear why the authors show the immunostainings in low magnification. Also, it is suggested to show gammaH2AX and YOYO1 in separate channels.

10. In the discussion section, the authors state, "The knockout of Aspm in neural progenitor cells increased apoptosis in all stages of cortical development compared to normal brains (Fig 4), but postnatal brains showed no differences from normal brains (data not shown)". Therefore, it is highly recommended that the authors include the postnatal data since it is an essential part of their discussion.

11. It is strongly suggested to add the supplemental material to the manuscript.

Reviewer #2: The manuscript by Tonosaki et al. provides evidences that Aspm (abnormal spindle-like microcephaly associated) is necessary to prevent excessive apoptosis in the developing brain cortex. Mutations in ASPM are among the most common causes of primary microcephaly in humans, and previous studies have focused on the analyzing mechanisms of Aspm in regulating cerebral cortical size. This manuscript originally shows that the elimination of Aspm in the developing telencephalon triggers an increase in cell death, suggesting a different (or additional) function from those previously described. The data presented supports the conclusion to a certain extent, and several major points should be addressed.

1. The Nes-Cre transgenic line used for generating cKOs is not described (information is missing in Methods). It is important to validate the genetic approach. The most widely used Nes-Cre Tg mouse (Tronche et al. Nat Genet 1999; JAX #003771) is known to be insufficient for directing recombination in early embryonic neural cortical progenitors (Liang, et al Biol Open 1:1200-3, 2012 and others), being Nes-Cre inactive in VZ or SVZ cells at E12.5 or even at E14.5. Assessing when Aspm deletion takes place is central because it might change conclusions of the study. For example, the statements “microcephaly associated with MCPH5 is not due to aberrant proliferation of neural progenitor cells during cortical development” (line 337, 207, others), “cerebral cortical reduction in Aspm cKO was exhibited at the late embryonic stage” (173, abstract), as well as discrepancies with previous reports, might arise from the inefficacy of deleting Aspm in VZ progenitors with the genetic approach used

Which was the Nes-Cre Tg line used? The efficacy of Cre recombination in the developing cortex should be tested (with conditional reporters) or cite the corresponding validations.

The downregulation of Aspm, at mRNA or protein levels, should be assessed in the Nes-Cre Aspm-flox/flox used in this study.

2. Some of the major findings should be validated with a complementary analysis in Aspm full knock out embryos (previously used by the authors of this manuscript, Fujimori et al 2014) or in Emx1-Cre; Aspm-flox (Jayaraman et al 2016).

3. Findings and differences with published work (Fish 2006, Jayaraman 2016) should be compared and discussed, taking into consideration animal models used, different ages or markers tested. As an example, upper layers thickness by P10 showed in cited works, is not evident here.

4. The authors propose a reduction in Tbr1+ thickness (line 190, Fig 1H-K). This would benefit from determining differences in cell numbers instead of thickness, in order to assert whether neurogenesis is affected, and considering the effect on thickness is mild.

Importantly, this reduction is not consistent with later results (Fig.3) where birth dating experiments at E12.5 do not show deep layer differences between WT and cKO.

5. In EdU/BrdU labeling experiments (Fig 3) it would be useful to also test for differences in number and distribution of EdU cells at a later timepoint (P5-10) when neurons have completed migration. The reduction in EdU cell numbers seen in the IZ but not CP, which suggests they are later born, would also be supported by Tbr1 co-staining. Authors should also consider adding an EdU pulse at E16.5 to solidly conclude which neurons are affected.

Additionally, in Fig 3 E-F, no EdU + cells are shown in quantification in bins 1-2 (or VZ/SVZ), but they are clearly evident in the images. Distinguishing between EdU cells remaining in VZ or migrating into IZ (with short chase times), coupled with the use of specific markers could help determine whether there is changes in neurogenic divisions. For this point, authors should consider similarities and/or differences with published work.

Minor

6. Examples of DNA damage in both Pax6+ and Pax6- cells are shown in Fig Suppl 4. Since numbers were not quantified, lines 314-315 should be revised. Quantifications of both cell type specific death and DNA damage would increase the robustness of the results.

7. It would be ideal to show the actual data points in the graphs together with mean +/- SEM.

8. Please, rephrase “..observed cortical reduction in the late neurogenesis of murine cortical development” Abstract, line 27.

9. Line 59: mutations in ASPM are a cause (not consequence) of mitotic aberrations during neurogenesis.

10. References for mouse lines are missing in Methods section.

11. Please indicate ages in Figures 1 F-H, 1E, 3 A-F where they are missing so that is easily interpreted.

12. Label the stainings close/within the panels in all figures. They are missing in Figs. 2, 3A-C, 5A-F and Fig.S2 A-F.

13. Line 339 should be modified. The authors show that there is no excessive or decreased proliferation but can’t exclude that there are other ways of aberrant division such as altered ratio of self-renewal vs. neurogenic divisions.

14. Please avoid the use of “cortical areas” as it may get confused with functional areas of the cortex (line 270). I suggest to remove the word “layers” (line 32, abstract)

15. I think it is more common to use embryonic instead of embryonal (176, 190 and others)

6. PLOS authors have the option to publish the peer review history of their article (what does this mean?). If published, this will include your full peer review and any attached files.

Reviewer #1: No

Reviewer #2: No

---

## [Author Response · Author response to Decision Letter 0]

8 Jul 2023

Response to reviewer #1’ comments

First of all, thank the reviewer #1 for reviewing our manuscript and for the valuable comments. We have tried to reply all the comments from the reviewer #1 and the details of the changes made are described as follows.

Major Comments:

Although most of the experiments suggested this, there are some data interpretation inconsistencies. For example, the authors mentioned that the reduced cortical thickness is due to early neurogenesis since the Tbr1-associated layer was the only one reduced. However, they do not show changes in cortical thickness at early neurogenesis, only at late neurogenesis. 

Furthermore, as the Tbr1 neurons are the first to be generated, how can they see cortical defects only at E16.5 and not at E12.5 or E13.5? Or, if the changes are only seen at E16.5, why are there no changes in Satb2 neurons?

I would greatly appreciate your important suggestions.

According to your suggestions, we prepared the supplemental figure (S6 Fig), which included the results about cortical thickness at E12.5, E14.5 and E16.5. Because the cortical layer at E12.5 was before the formation of the cortical plate, we evaluated it as preplate thickness, which showed immunoreactivity for Tbr1.

In terms of the cortical wall, which is total thickness of cerebral cortex, there was a significant difference at late state of murine brain development. However, the cortical thickness, which is the preplate or the cortical plate, did not show the significant difference at all ages. This result revealed that early cortical thickness did not change. Then we have thought that it was due to the loss of neurons or the reduction in generating neurons. 

Page 9, line 196

We further analyzed how the changes were related to the layer construction of the cerebral cortex at E16.5. First, there was a significant decrease in the thickness of the VZ in Aspm cKO (Figs 1C’, 1E and 1F, and S8 Fig). Although the thickness of IZ or CP showed a slight decrease, the significance did not reach the level of significance as compared to WT (Figs 1C’, 1E and 1F, and S6 Fig).

The CP is composed of six layers, and each layer consists of neurons with different birthdates. Therefore, we examined the CP in more detail using specific markers for each cortical layer. In the mouse brains lacking Aspm, the number of cells immunoreactive for the upper layer marker (Satb2-positive, II/III and IV layer), the middle layer marker (Ctip2-positive, layer V) and for the deep layer marker (Tbr1-positive, layer VI) showed a slight decrease, but no significant differences, when compared to those of the WT mice, (Figs 1G-1L), suggesting that a normal cytoarchitecture of the cerebral cortex was formed even with a loss of Aspm. 

These findings suggest that the thinning of the cortex in late cortical formation might be due to poor neurogenesis or an excess loss of neurons at all embryonal stages.　

I would greatly appreciate your important suggestions.

As mentioned above and in the Fig 1D, the thickness of cortical wall showed no significant difference until E14.5, and reduction in thickness was observed at E16.5. In addition, we showed that there was no difference in the number of Ctip2, Satb2 and Tbr1-positive neurons at E16.5 (Fig 1G-L).

Thus, we thought that the excess loss of neurons might continuously occur at all embryonal stages. We have demonstrated the significant increase in apoptosis in the developing cerebral cortex at all embryonal stages, however; apoptosis in the VZ was higher at E12.5 and E14,5 than at E16.5 as shown in the Fig 4, it was possible that loss of the neuronal progenitor cells might affect progression of corticogenesis. 

Page 10, line 231

Loss of Aspm led to an increase in apoptosis of both NPCs and postmitotic neurons

We conducted assays to determine whether the occurrence of apoptosis was related to the cortical thinning observed in association with the loss of Aspm (Fig 4). In the TUNEL assay for measuring apoptosis, TUNEL-positive cells were distributed in the VZ and PP at E12.5 (Figs 4A, 4B and 4J) and in the VZ, IZ and CP at E14.5 and E16.5 (Figs 4C-4F and 4J). The TUNEL-positive cells were significantly increased in the developing cortex in the cKO mice throughout corticogenesis (Fig 4G). The numerical density of the TUNEL-positive cells in the developing cortex areas (TUNEL index) decreased as the embryonal stage proceeded (Figs 4H and 4I). Furthermore, the TUNEL-index was significantly higher in the VZ at E12.5 and 14.5, but not at E16.5 in the cKO mice as compared to WT mice (Fig 4J). Thus, we found an increased number of apoptosis instances in the developing cortex obtained from Aspm cKO mice and the apoptosis was higher in the VZ at the early to the middle phase of the corticogenesis, compared with the later corticogenesis.

In order to confirm the features of the apoptosis, we employed nuclear morphology, including nuclear condensation and DNA fragmentation, and the molecular signatures were evaluated by immunohistochemistry for CC3 (S2 Fig). The CC3-positive cells and/or nuclear condensation were significantly increased in the cKO mice at E12.5, distributed in the VZ and PP. The CC3-positive cells and/or nuclear condensation were significantly increased at E14.5 and E16.5 as well. Furthermore, a significant increase in the numerical density of CC3-positive cells (CC3 index) was revealed in the VZ at E14.5 and in the IZ at E16.5 in samples obtained from the developing cortex of the cKO mice. The numerical density of nuclear condensation (nuclear index) showed a significant increase in the VZ at E14.5 in cKO mice.

Next, we conducted tests to determine whether the apoptosis was induced in neural progenitor cells, intermediate progenitors, or neurons by the loss of Aspm (S3 Fig). TUNEL-positive cells expressed the neuronal markers NeuN, Tuj1, and DCX (S3 A-C, A’-C’ Figs), but not the APs marker Pax6 or BPs marker Tbr2 (S3 D-E, D’-E’ Figs). In CC3-positive cells, the expression was consistent with the neural progenitor cell marker Nestin (S3 F and F’ Figs). 

Judging from these results, we conclude that Aspm expression is required for the maintenance of neural progenitor cells and neurons and plays a critical role during neurogenesis.

3.Also, the authors did not cite or discuss important published articles such as Johnson et al., 2018. Nature or Williams et al., 2015. Development, where they mentioned that loss of Aspm is also associated with apoptosis and important for brain development. 

I would greatly appreciate your important suggestions.

As indicated, we have cited in the Discussion.

Page 19, line 431 

Johnson MB et al. [22] published an excellent study showing that Aspm KO ferrets, a gyrencephalic animal, exhibited severe microcephaly, reflecting reduced cortical surface area without any significant change in cortical thickness, as in human patients. The mutant ferret fetal cortex displayed a massive premature displacement of ventricular radial glial cells (VRG), the most undifferentiated cell type, to the outer subventricular zone (OSVZ), and a more differentiated progenitor, resulting in a loss of "cortical units". This finding is similar to that in another previous report [21]. Unfortunately, we did not find remarkable displaced progenitors, or severe microcephaly in the present Aspm cKO brains, which is most likely because of that mice, agyrencephalic animal, have few OSVG, Tbr2-immunoreactive BPs.

Page 17, line 398 and 414 

It has been reported that the deletion of Aspm induced prematurely exit of the cell cycle and early differentiation of the cerebellar granule neuron progenitors, resulting in a decrease of terminally differentiated granule neurons in the internal granule layer [18].

Aspm-deficient cerebellar granule neuron progenitors show impaired mitotic progression, altered patterns of division orientation and differentiation, and increased DNA damage, which causes progenitor attrition through apoptosis [18]. Similar mechanisms might be involved in the abnormal corticogenesis in our Aspm cKO.

Comments:

1. Fig. 1E does not explain how VZ/, IZ, or CP was defined.

I would greatly appreciate your important suggestions.

We defined the VZ, IZ, and CP as follows: The CP was defined based on DAPI staining and the VZ was defined based on SOX2 staining. The IZ was defined as the value obtained by subtracting the CP and VZ from the cortical wall, which was mesured DAPI staininig. The thickness was as average of three points (left, center and right within 250 µm-wide dorsal cortex regions, n = 3 brains) because each layer thickness was not uniform.

We have remade the Figure 1, which included the image to show a definition of the VZ, IZ and CP using the SOX2/DAPI staining.

Page 10, line 217

(E) SOX2 and DAPI stained image of E16.5 mouse cerebral cortex. The two head arrows indicate the CP, VZ and cortical wall. (F) Quantification of cortical wall and the CP region were defined based on DAPI staining and that of VZ was defined based on SOX2 staining. The IZ was defined as the value obtained by subtracting the CP and VZ from cortical wall. (VZ, WT 85.54 ± 2.2 μm, cKO 70.59 ± 2.1 μm, P = 0.0104; IZ, WT　229.29 ± 4.9 μm, cKO 210.55 ± 9.3 μm, P = 0.1490; CP, WT 159.51 ± 6.4 μm, cKO 140.03 ± 9.0 μm, P = 0.0908, Student’s t-test, values represent mean ± SEM, n = 3 mice per genotype) (G-I) Representative images of the WT and Aspm cKO Satb2, Ctip2 and Tbr1, respectively. Scale bar represents 50 µm. (J-L) Quantification of the cell number of the upper layer marker Satb2, the middle layer maker Ctip2 and the deep layer marker Tbr1, respectively, per unit area G-I (250μm radial columns). (J: WT 314.0 ± 23.4 cells, cKO 266.7 ± 36.2 cells, P = 0.3336, K: WT 392.7 ± 19.3 cells, cKO 327.7 ± 26.8 cells, P = 0.1205, L: WT 416.0 ± 14.7 cells, cKO 377.3 ± 17.9 cells, P = 0.1702, Student’s t-test, values represent mean ± SEM, n = 3 mice per genotype, ns: not significant). 

2. In Fig. 1F-K it is not explained how the authors defined the areas measured. For example, for SATB2, several cells spread across the cortex are SATB2 positive. Then, what parameter is used by the authors to establish the right quantification area? Quantifying the number of SATB2, CTIP2 and TBR1 positive cells would be more appropriate. Also, the authors did not state the age of the samples analyzed. If the samples are embryonic, how can they define specific cortical layers since they have not been adequately established yet?

I would greatly appreciate your important suggestions.

1) As suggested, we have changed the graphs Fig 1I-K to the number of Satb2, Ctip2 and Tbr1 positive cells, respectively.

2) We apologize for the unclear presentation. The samples in the Fig1G-I were images of embryonal day 16.5 (E16.5) brains in order to analysis the cortical layer construction refer to the Fig 1C. We have added the labels refer to the embryonal date to all figures in order to easily recognize the embryonic date.

3) As your suggestion, cortical layers have not been adequately established yet at the embryonal stage. However, the Satb2, Ctip2 and Tbr1 were used for analyzing the cortical layer thickness in the E16.5 Aspm-null cortex in the published report by Fujimori A et al (Brain & Development 36 (2014) 661–669). Therefore, we also conducted the immunostaining such as the Satb2, Ctip2 and Tbr1 to confirm the difference in the layer formation between WT and cKO.

Therefore, we have added to the co-staining image of SOX2 and DAPI to Fig 1(Fig 1E). According to the change, we have changed Fig 1 and Figure legend as follows: 

Page 9, line 210

Fig 1. Embryonal brain of Aspm cKO

(A-C) Representative DAPI stained images of the embryonal cortices of the WT and Aspm cKO mice (NesCre;Aspmflox/flox) at E12.5, E14.5 and E16.5. The scale bars represent 500 μm. (A’-C’) Magnification of the boxed area in (A-C). The scale bars represent 50 μm. (D) Quantification of the thickness per unit area (A’-C’) (250 μm radial columns) *: P < 0.05 (E12.5: WT 151.1 ± 2.0 μm, cKO 155.1 ± 6.5 μm, P = 0.5922, E14.5: WT 238.0 ± 13.5 μm, cKO 226.9 ± 5.8 μm, P = 0.4913, E16.5: WT 482.4 ± 7.5 μm, cKO 427.1 ± 15.5 μm, P = 0.0325, Student’s t-test, values represent mean ± SEM, n = 3 mice per genotype, ns: not significant) (E) SOX2 and DAPI stained image of E16.5 mouse cerebral cortex. The two head arrows indicate the CP, VZ and cortical wall. (F) Quantification of cortical wall and the CP region were defined based on DAPI staining and that of VZ was defined based on SOX2 staining. The IZ was defined as the value obtained by subtracting the CP and VZ from cortical wall. (VZ, WT 85.54 ± 2.2 μm, cKO 70.59 ± 2.1 μm, P = 0.0104; IZ, WT　229.29 ± 4.9 μm, cKO 210.55 ± 9.3 μm, P = 0.1490; CP, WT 159.51 ± 6.4 μm, cKO 140.03 ± 9.0 μm, P = 0.0908, Student’s t-test, values represent mean ± SEM, n = 3 mice per genotype) (G-I) Representative images of the WT and Aspm cKO Satb2, Ctip2 and Tbr1, respectively. Scale bar represents 50 µm. (J-L) Quantification of the cell number of the upper layer marker Satb2, the middle layer maker Ctip2 and the deep layer marker Tbr1, respectively, per unit area G-I (250μm radial columns). (J: WT 314.0 ± 23.4 cells, cKO 266.7 ± 36.2 cells, P = 0.3336, K: WT 392.7 ± 19.3 cells, cKO 327.7 ± 26.8 cells, P = 0.1205, L: WT 416.0 ± 14.7 cells, cKO 377.3 ± 17.9 cells, P = 0.1702, Student’s t-test, values represent mean ± SEM, n = 3 mice per genotype, ns: not significant).

4) According to the change, we have changed the description in the Results as follows:

Page 9, line 188

Cerebral cortical reduction in Aspm cKO was exhibited at the late embryonal stage

First, we generated brain specific Aspm knockout mouse (NesCre;Aspmflox/flox, Aspm cKO) (S1 Fig) in order to examine whether the morphology of the embryonal cerebral cortex was affected by the loss of Aspm. Cortical thickness did not significantly change between the WT and cKO mice at E12.5 and E14.5 (E12.5, WT 151.1 ± 2.0 µm, cKO 155.1 ± 6.4 µm, P = 0.5922; E14.5, WT 238.0 ± 13.5 µm, cKO 227.0 ± 5.8 µm, P = 0.4913). However, a morphological difference was observed at E16.5, with a decrease in cortical thickness (WT 482.4 ± 7.5 µm, cKO 427.1 ± 15.5 µm, P = 0.0325) (Figs 1A-1D).

We further analyzed how the changes were related to the layer construction of the cerebral cortex at E16.5. First, there was a significant decrease in the thickness of the VZ in Aspm cKO (Figs 1C’, 1E and 1F, and S8 Fig). Although the thickness of IZ or CP showed a slight decrease, the significance did not reach the level of significance as compared to WT (Figs 1C’, 1E and 1F, and S6 Fig).

The CP is composed of six layers, and each layer consists of neurons with different birthdates. Therefore, we examined the CP in more detail using specific markers for each cortical layer. In the mouse brains lacking Aspm, the number of cells immunoreactive for the upper layer marker (Satb2-positive, II/III and IV layer), the middle layer marker (Ctip2-positive, layer V) and for the deep layer marker (Tbr1-positive, layer VI) showed a slight decrease, but no significant differences, when compared to those of the WT mice, (Figs 1G-1L), suggesting that a normal cytoarchitecture of the cerebral cortex was formed even with a loss of Aspm. 

These findings suggest that the thinning of the cortex in late cortical formation might be due to poor neurogenesis or an excess loss of neurons at all embryonal stages.　

3. The authors mentioned, "Since the deep layer neurons are composed of early-born neurons, these findings suggest that the thinning of the cortex in late cortical formation might be due to poor neurogenesis at early embryonal stages". If only early neurogenesis is affected, how later is neurogenesis not? Depleting NPCs at early stages should also affect later neurogenesis as the NPC pool is reduced. Also, it is possible that the loss of ASPM only affects direct neurogenesis but not indirect neurogenesis. However, the authors did not test this possibility.

I would greatly appreciate your important suggestions.

In order to clarify the point, we have prepared additional supplemental figure (S8 Fig) to show the SOX2 staining for assessment of the thickness of the ventricular zone as NPC pool size at E12.5, E14.5 and E16.5. According to this figure, there was a significant difference between WT and cKO only at E16.5, suggesting that the NPC pool was reduced at the late stage. 

We described in Results as follows.

Page 15, line 345

In addition, the thickness of VZ and the number of NPCs, immunoreactive for PAX6, showed a decrease at E16.5 in cKO compared to WT (S8 Fig).

Altogether, these results suggest that the loss of Aspm causes an increase in the number of cells with DNA damage in developing murine cortex, resulting in a reduction of VZ at the later corticogenesis stage.

As your suggestion, we did not precisely evaluate that the loss of Aspm affected the direct neurogenesis and/or indirect neurogenesis. The present data is supposed to show the effects on the direct neurogenesis, which is most likely because of that mice, agyrencephalic animal, have fewer indirect neurogenesis. 

We described in Discussion as follows:

Page 19, line 431

Johnson MB et al. [22] published an excellent study showing that Aspm KO ferrets, a gyrencephalic animal, exhibited severe microcephaly, reflecting reduced cortical surface area without any significant change in cortical thickness, as in human patients. The mutant ferret fetal cortex displayed a massive premature displacement of ventricular radial glial cells (VRG), the most undifferentiated cell type, to the outer subventricular zone (OSVZ), and a more differentiated progenitor, resulting in a loss of "cortical units". This finding is similar to that in another previous report [21]. Unfortunately, we did not find remarkable displaced progenitors, or severe microcephaly in the present Aspm cKO brains, which is most likely because of that mice, agyrencephalic animal, have few OSVG, Tbr2-immunoreactive BPs.

4.The authors mentioned that "Immunofluorescence studies for pHH3, which is the mitotic marker, revealed no significant differences in the abundance of apical progenitor cells (APs) and basal progenitor cells (BPs) between the WT and Aspm cKO mice (Figs 2A and 2B)". How did the authors determine apical or basal progenitors? The authors assume the nature of these progenitors based on their spatial location. However, it is highly recommended that these experiments be performed using proper cell markers such as Pax6 or Tbr2 and colocalization studies.

I would greatly appreciate your important suggestions.

As suggested, we did not precisely detect apical progenitor cells and basal progenitor cells using specific immunostaining such as Pax6 and Tbr2. However, according to the published report by Arai Y et al (Nat Commun. 2011, ref 17), apical progenitor cells and basal progenitor cells were determined based on the spatial location. Furthermore we did not determine APs and BPs using co-staining such as pHH3/Pax6 and pHH3/Tbr2. However, as mentioned above, the determination of APs and BP based on the spatial location in the published report. So, we thought that it was useful to evaluate APs and BPs based on the spatial location of pHH3 positive cells.

Thus, we have detected them based on the spatial localization and described as follows:

Page 10, line 233

In order to identify specific neural progenitors, apical progenitor cells (APs) were defined as the phospho-histone H3 (pHH3) positive cells located beneath the surface of the lateral ventricle, and basal progenitor cells (BPs) were defined as the pHH3 positive cells localized in the basal region of the ventricular zone (Fig 2) [17] Immunofluorescence studies for pHH3, which is the mitotic marker, revealed no significant differences in the abundance of APs and BPs between the WT and Aspm cKO mice (Figs 2A and 2B).

5.In Fig. 2C-D, it is unclear whether the absence of proliferative disruption is specific to a particular progenitor population since the authors did not use any cell-specific marker. Therefore, it is highly recommended that these experiments be performed using proper cell markers such as Pax6 or Tbr2 since the authors state that only early neurogenesis is affected.

I would greatly appreciate your important suggestions.

I agree with you that these experiments should be performed using proper cell markers such as Pax6 or Tbr2 in order to assess whether direct or indirect neurogenesis was more affected in the present mice brain. However; we suggest that a few Tbr2-immunoreactive BPs, OSVG, exist in mice brain, agyrencephalic animal, and proliferation of neuronal proliferation is unlikely involved in reduction of cortical neurons in Aspm cKO. 

6.The authors mentioned, "Pregnant mice were given a single dose of EdU at E12.5 or E14.5, and two days later (i.e., E14.5 or E16.5, respectively), the abundance of EdU-positive neurons was measured. Most of the cells, EdU-labeled at E12.5, were observed in the superficial area corresponding to the CP, and EdU-positive cells were found in postmitotic neurons showing Tbr1-immunoreactivity". However, TBR1 immunostaining was not performed. So then, how can the authors ensure that EdU staining corresponds to postmitotic neurons? Also, the authors made similar conclusions on Fig. 3C-F but did not perform immunostainings for specific neuronal populations such as Satb2, Ctip2 or Tbr1.

I would greatly appreciate your important suggestions.

According to your suggestions, in terms of the birthdating assay at early to middle stage, we have confirmed co-immunostaining of EdU and specific neural maker such as Tbr1 and remade the Fig 3, which includes co-immunostaining for Tbr1, EdU and DAPI images in order to show that the EdU positive cells were postmitotic neurons in cortical plate at E14.5 (Figs 3A and B).

We described as follows: 

Page 12, line 263

Most of the cells, EdU-labeled at E12.5, were observed in the superficial area corresponding to the CP, and EdU-positive cells were found in postmitotic neurons showing Tbr1-immunoreactivity. The E12.5-labeled neurons showed no significant differences between the WT and cKO mice (Figs 3A and 3B).

Regarding the double staining ratio, we made supplemental figure (S7 Fig), which revealed that Tbr1/EdU was about 8.7% in WT and 9.9% in cKO, that of Ctip2/EdU was about 3.4% in WT and about 3.4% in cKO, and that of Satb2/EdU was about 76.6% in WT and 86.9% in cKO, respectively (S7 Fig). 

We added the description as follows:

Page 12, line 266.

In contrast, the number of cells labeled at E14.5 was significantly reduced, with most of them scattered from the IZ to the CP, in the cKO mice. There was a significant reduction in the number of EdU-labeled cells distributed in the IZ, but not in the VZ and CP (Figs 3C-3F). Although most of the EdU-labeled cells showed immunoreactivity for Satb2, there were no significant differences in ratio of Tbr1+/EdU+, Ctip2+/EdU+, and Satb2+/EdU+ cells in the cerebral wall between the WT and Aspm cKO brains (S7 Fig). 

These findings suggested that the loss of Aspm induced a reduction in the number of mid to later born neurons without any aberrant expression of cerebral layer-specific transcription factors in the developing cortex.

As suggested, we have remeasured for the number of EdU-positive cells and remade the Fig 3, which included quantitative results about VZ/SVZ (bin1/bin2). Furthermore, it was performed Edu+/Tbr1+ cells because neurons at the lower CP and subplate (SP) were Tbr1-positive and the border between CP and SP was distinguishable to indicate the cortical layer (CP, IZ and VZ/SVZ) and bin number (Bin1-10).

We also have remade the figure legend as follows: 

Page 12, line 276

Fig 3. Neurogenesis during murine corticogenesis. 

(A, C) Co-staining images of Tbr1 (green), EdU (red) and DAPI (blue) in the WT and Aspm cKO mice at E14.5 (A) and E16.5 (C) cortices two days after the EdU injections (E12.5 and E14.5, respectively). Scale bar: 50 µm. (B, D) The number of EdU-positive postmitotic cells per 250 µm. *P < 0.05 (B: WT 50.67 ± 3.5, cKO 54.3 ± 7.9, P = 0.6923, D: WT 228.7 ± 6.9, cKO 174.7 ± 11.4, P = 0.0154, Student’s t-test, values represent mean ± SEM, n = 3 mice per genotype) (E) Histogram of the EdU-positive cells, which were shown in C. (bins × genotype interaction F(8, 36) = 2.893 P = 0.0135; two-way ANOVA with Tukey’s multiple comparisons test; **P < 0.01, values represent mean ± SEM, n = 3 mice per genotype). (F) Comparison of the distribution of the EdU-positive cells among the cortical layers (cortical layers × genotype interaction F(2, 12) = 27.82 P < 0.0001; two-way ANOVA with Tukey’s multiple comparisons test; ****P < 0.0001, ns: not significant, values represent mean ± SEM, n = 3 mice per genotype)

7. The authors do not have data to support the following statement: "These findings suggest that the loss of Aspm induced a reduction in the number of mid-born neurons under radial migration" since they did not perform neuronal migration experiments.

I would greatly appreciate your important suggestions.

As your suggestions, we did not perform precise migration assay in the present study. However, most of postmitotic neurons in the IZ are young neurons under radial migration in the developing cerebral cortex. 

We described in Discussion as follows:

Page 17, line 391.

Therefore, it is highly likely that cell loss might occur not only in NPCs, but also in postmitotic neurons, i.e., neuroblasts in the VZ/SVZ, migrating neurons in the IZ or immature neurons after reaching the CP.

8.Fig. 4A-H are not cited in the text.

I would greatly appreciate your important suggestions.

We have added the citation in the text appropriately. 

Page 13, line 290.

We conducted assays to determine whether the occurrence of apoptosis was related to the cortical thinning observed in association with the loss of Aspm (Fig 4). In the TUNEL assay for measuring apoptosis, TUNEL-positive cells were distributed in the VZ and PP at E12.5 (Figs 4A, 4B and 4J) and in the VZ, IZ and CP at E14.5 and E16.5 (Figs 4C-4F and 4J). The TUNEL-positive cells were significantly increased in the developing cortex in the cKO mice throughout corticogenesis (Fig 4G). The numerical density of the TUNEL-positive cells in the developing cortex areas (TUNEL index) decreased as the embryonal stage proceeded (Figs 4H and 4I). Furthermore, the TUNEL-index was significantly higher in the VZ at E12.5 and 14.5, but not at E16.5 in the cKO mice as compared to WT mice (Fig 4J). Thus, we found an increased number of apoptosis instances in the developing cortex obtained from Aspm cKO mice and the apoptosis was higher in the VZ at the early to the middle phase of the corticogenesis, compared with the later corticogenesis.

9.In Figure 5, it is unclear why the authors show the immunostainings in low magnification. Also, it is suggested to show gammaH2AX and YOYO1 in separate channels.

I would greatly appreciate your important suggestions.

As suggested, we have remade Fig.5 which showed at higher magnification. We showed co-immunostaining images of γH2AX and YOYO1 as well as separated images. According to the change, we have changed Fig 5 and the Figure legend as follows:

9)-1 We have changed the Fig 5.

Page 15, line 351.

Fig 5. Immunohistochemistry for γH2AX

(A-F) Representative immunofluorescence images (γH2AX (red) and YOYO1 (blue) of embryonal cortex in the WT and Aspm cKO mice at E12.5 (A, B), E14.5 (C, D) and E16.5 (E, F). The number of γH2AX-positive cells were counted in the developing cortex. Arrows indicate γH2AX-positive cells (red). Scale bars represent 500 μm. (G) The number of γH2AX positive cells in cerebral cortex (E12.5: WT1.00 ± 0.33 cells, cKO 3.89 ± 0.80 cells P = 0.0294, E14.5 WT 4.67 ± 0.69 cells, cKO 8.00 ± 0.39 cells P = 0.0137, E16.5: WT 3.11 ± 1.87 cells, cKO 4.67 ± 0.88 cells P = 0.4936, Student’s t-test, *P < 0.05, values represent mean ± SEM, n = 3 mice per genotype) (H) The area of cerebral certex which cell number was counted. (E12.5: WT 0.12 ± 0.0 mm2, cKO 0.16 ± 0.01 mm2, P = 0.0027, E14.5: WT 0.31 ± 0.04 mm2, cKO 0.30 ± 0.01 mm2, P = 0.8369, E16.5: WT 0.50 ± 0.04 mm2, cKO 0.48 ± 0.01 mm2, P = 0.5989, Student’s t-test, **P < 0.01, values represent mean ± SEM, n = 3 mice per genotype) (I) γH2AX index (Num/Area) is the number of γH2AX-positive cells (G) in a cerebral cortex divided by the unit area (H). (E12.5: WT 8.27 ± 3.04 cells/mm2, cKO 23.89 ± 4.79, P = 0.0521, E14.5: WT 15.87 ± 4.56 cells/mm2, cKO 26.32 ± 0.94, P = 0.0880, E16.5: WT 6.21 ± 3.93 cells/mm2, cKO 9.78 ± 1.62, P = 0.4484, ns: not significant, values represent mean ± SEM, n = 3 mice per genotype) (J) Distribution of γH2AX index in the developing cortex divided into the VZ, IZ, PP and/or CP. (E12.5: cortical layers × genotype interaction F(1, 8) = 3.387 P = 0.1030, E14.5: cortical layers × genotype interaction F(2, 12) = 0.2642 P = 0.7722, E16.5: cortical layers × genotype interaction F(2, 12) = 0.4965 P = 0.6206; two-way ANOVA with Tukey’s multiple comparisons test; *P < 0.05, ns: not significant, n = 3 mice per genotype) 

We have added the description in the Method as follows: 

Page 7, line 167.

In order to measure the γH2AX positive cells, we used ImageJ plugin “Analyze Particles” and selected a signal size within 3.00－200.00 to remove the particles that were too small or big, and thus regarded as false positive signals.

10.In the discussion section, the authors state, "The knockout of Aspm in neural progenitor cells increased apoptosis in all stages of cortical development compared to normal brains (Fig 4), but postnatal brains showed no differences from normal brains (data not shown)". Therefore, it is highly recommended that the authors include the postnatal data since it is an essential part of their discussion.

I would greatly appreciate your important suggestions.

As suggested, we made the Supplemental Figure S5 Fig to show the TUNEL staining images of P7 brains as postnatal brains. Although we have evaluated the staining qualitatively, TUNEL-positive cells at the P7 brain were apparently lower than in embyonal brains. 

We added the description as follows:

Page 13, line 293.

Although the TUNEL-positive cells were significantly increased in the developing cortex in the cKO mice throughout corticogenesis (Fig 4G), postnatal brains showed no differences from WT brains (S5 Fig).

10)-2 We have prepared the Supplemental Figure S5 Fig.

We also added the supplemental figure legend (S5 Fig legend) as follows: 

Page 27, line 600.

S5 Fig. TUNEL staining of postnatal mouse brain at P7 

(A-B) representative TUNEL staining images of WT and Aspm cKO postnatal brains at P7. Scale bars represent 1 mm. (A’-B’) Higher magnification of the boxed area in A and B, respectively. Arrowheads indicated TUNEL-positive cells.

We added the sentense in the Method as follows

Page 8, line 175.

For P7 brains (S5 Fig), a Nikon Ti Eclipse confocal microscope were used, and the images were processed with ImageJ/Fiji to adjust color and contrast. Whole brain images were obtained by tiling and stitching (6×6) the images captured with a 20× lens (2048×2048 pix). The high magnification images were cropped from the whole brain images.

11.It is strongly suggested to add the supplemental material to the manuscript.

I would greatly appreciate your important suggestions.

We described all precise procedures in Materials and Methods of the manuscript even for supplemental figures as well. As mentioned above, the sentence “data not shown” was changed to “S5 Fig” in order to show the actual result.

Response to reviewer #2’ comments

First of all, I would greatly appreciate reviewer’s suggestions and comments on our manuscript. We have tried to revise the former manuscript according to valuable comments.

Refer to Materials

1. The Nes-Cre transgenic line used for generating cKOs is not described (information is missing in Methods). It is important to validate the genetic approach. The most widely used Nes-Cre Tg mouse (Tronche et al. Nat Genet 1999; JAX #003771) is known to be insufficient for directing recombination in early embryonic neural cortical progenitors (Liang, et al Biol Open 1:1200-3, 2012 and others), being Nes-Cre inactive in VZ or SVZ cells at E12.5 or even at E14.5. Assessing when Aspm deletion takes place is central because it might change conclusions of the study. For example, the statements “microcephaly associated with MCPH5 is not due to aberrant proliferation of neural progenitor cells during cortical development” (line 337, 207, others), “cerebral cortical reduction in Aspm cKO was exhibited at the late embryonic stage” (173, abstract), as well as discrepancies with previous reports, might arise from the inefficacy of deleting Aspm in VZ progenitors with the genetic approach used.

1-1) Which was the Nes-Cre Tg line used? The efficacy of Cre recombination in the developing cortex should be tested (with conditional reporters) or cite the corresponding validations.

I would greatly appreciate your important suggestions. 

We used Nestin-cre (C57BL/6-Tg(Nes-Cre)1Kag, Center for Animal Resources and Development (CARD), CARD-ID 650) mice [Isaka, F. et al. Eur.J. Neuroscience 11. 2582-2588, 1999] It was reported to be sufficient for directing recombination in early embryonic neural cortical progenitors.

Since we did not find any changes in proliferation of the neuronal progenitor cells from E12.5 to E16.5, namely during active proliferation stage in corticogenesis, but found increased apoptosis in above stage, we described “the microcephaly associated with MCPH5 might be aggravated by increased apoptosis in NPCs and postmitotic neurons, but not by aberrant proliferation of NPCs during cortical development. (Page 17, line 374, Page 11, 244, others). The loss of postmitotic cells, including neurons and progenitor cells resulted in “cerebral cortical reduction in Aspm cKO was at the late embryonic stage.

We added descriptions in Material and Methods as follows

Page 4, line 81. 

In order to obtain NesCre;Aspmflox/+ mice, Nestin-cre (C57BL/6-Tg(Nes-Cre)1Kag, Center for Animal Resources and Development (CARD), CARD-ID 650) mice [16] were mated with Aspmflox/flox mice [9]. Brain specific Aspm knockout mice (NesCre;Aspmflox/flox, Aspm cKO) were generated by mating Aspmflox/flox mice and NesCre;Aspmflox/+ mice and detected by genotyping.

In addition, we showed a loss of Aspm protein at E14.5 (S1 Fig).

Page 26, line 553. 

S1 Fig. Aspm cKO and wild type mice

This figure shows the Aspm cKO and wild type mice used in this study (see Materials and Methods). (A) The mouse Aspm gene targeting construct (B) Genotyping of the conditional Aspm knockout mouse (C) Immunostaining for Aspm (red), γTub (centrosome; green) and DNA (blue) in the NPCs of the wild type and conditional knockout (NesCre;Aspmf/f) E14.5 mice brains.

1-1. The downregulation of Aspm, at mRNA or protein levels, should be assessed in the Nes-Cre Aspm-flox/flox used in this study.

I would greatly appreciate your critical suggestions.

We showed a loss of Aspm protein in the ventricular stem cells of Aspm cKO mice brains at E14.5 by

immunohistochemistry (S1-C Fig).

S1 Fig. Aspm cKO and wildtype mouse

Page 26, line 553.

S1 Fig. Aspm cKO and wild type mice

This figure shows the Aspm cKO and wild type mice used in this study (see Materials and Methods). (A) The mouse Aspm gene targeting construct (B) Genotyping of the conditional Aspm knockout mouse (C) Immunostaining for Aspm (red), γTub (centrosome; green) and DNA (blue) in the NPCs of the wild type and conditional knockout (NesCre;Aspmf/f) E14.5 mice brains.

Refer to Materials 

2. Some of the major findings should be validated with a complementary analysis in Aspm full knock out embryos (previously used by the authors of this manuscript, Fujimori et al 2014) or in Emx1-Cre; Aspm-flox (Jayaraman et al 2016).

I would greatly appreciate your critical suggestions.

Aspm full knock out embryos (previously used by the authors of this manuscript, Fujimori et al 2014) were CAG-CreTg and different form the present mice. Furthermore, they did not analyze precisely embryonic brains. 

Jayaraman et al. reported that loss of Wdr62, Aspm or both results in expansion of basal progenitors beyond the germinal zones at mid-neurogenesis, suggesting regulation of cell fate leading to precocious formation of Tbr2+ cells rather than merely mitotic progression in Emx1-Cre; Aspm-flox (Jayaraman et al 2016). In addition, they showed little-to-no apoptosis in the Aspm-/- mice brains.

We did not find remarkable dislamination of basal progenitor cells beyond the germinal zones at E14.5, using birthdate analyses (Fig.3). Furthermore, the number of apoptotic cells showed significantly higher in all stages of corticogenesis in our Aspm cKO mice brains, suggesting the microcephaly might be followed by cell loss via apoptosis rather than mitotic progression or cell fate dysregulation. The reasons for the different results are unknown but might be derived from the differences in Aspm KO mice, and analyzed methods. 

We added in Discussion as follows:

Page 18, line 418.

Jayaraman et al. [21] reported that loss of Wdr62, Aspm or both impairs centriole duplication, correlating with the severity of microcephaly, and leads to a loss of centrosomes and cilia in the developing brain, and accordingly, microcephaly has been described as a ‘centriolopathy’. They showed an expansion of basal progenitors beyond the germinal zones at mid-neurogenesis, suggesting regulation of cell fate leading to precocious formation of basal progenitors rather than merely mitotic progression in Emx1-Cre; Aspm-flox. In addition, they showed little-to-no apoptosis in the Aspm-/- mice brains. We did not find any remarkable delamination of basal progenitor cells beyond the germinal zones at E14.5, using birthdate analyses (Fig 3). Furthermore, the number of apoptotic cells was significantly higher in all stages of corticogenesis in our Aspm cKO mice brains (Figs 4, 5 and S2 Fig), suggesting that the microcephaly might be followed by cell loss via apoptosis rather than proliferation and/or cell fate dysregulation. The reasons for these different results are unknown, but they might have been due to differences in the Aspm KO mice and experiment methods employed, including the ages or markers analyzed. 

Refer to discussion

3. Findings and differences with published work (Fish 2006, Jayaraman 2016) should be compared and discussed, taking into consideration animal models used, different ages or markers tested. As an example, upper layers thickness by P10 showed in cited works, is not evident here.

I would greatly appreciate your critical suggestions.

We added in Discussion as follows:

Page 17, line 386.

Fish et al. [13] reported that Aspm-downregulation, using RNA interference, induced the alteration in the cleavage plane orientation in neuroepithelial cells of embryonal telencephalon, followed by a reduction in the number of neuroepithelial progenitor cells and microcephaly. In the present study, we did not find any aberrant cleavage plane orientation in the neuroepithelial cells and the mitosis of the NPCs was not affected in the cortex of the cKO mice when we focused on the S and M phase of the cell cycle (Fig 2). Therefore, it is highly likely that cell loss might occur not only in NPCs, but also in postmitotic neurons, i.e., neuroblasts in the VZ/SVZ, migrating neurons in the IZ or immature neurons after reaching the CP. These observations suggest that this cell loss could play some role in building the normal cortical architecture, except for uniform thinning, as observed in the cKO mice.

Page 18, line 418.

Jayaraman et al. [21] reported that loss of Wdr62, Aspm or both impairs centriole duplication, correlating with the severity of microcephaly, and leads to a loss of centrosomes and cilia in the developing brain, and accordingly, microcephaly has been described as a ‘centriolopathy’. They showed an expansion of basal progenitors beyond the germinal zones at mid-neurogenesis, suggesting regulation of cell fate leading to precocious formation of basal progenitors rather than merely mitotic progression in Emx1-Cre; Aspm-flox. In addition, they showed little-to-no apoptosis in the Aspm-/- mice brains. We did not find any remarkable delamination of basal progenitor cells beyond the germinal zones at E14.5, using birthdate analyses (Fig 3). Furthermore, the number of apoptotic cells was significantly higher in all stages of corticogenesis in our Aspm cKO mice brains (Figs 4, 5 and S2 Fig), suggesting that the microcephaly might be followed by cell loss via apoptosis rather than proliferation and/or cell fate dysregulation. The reasons for these different results are unknown, but they might have been due to differences in the Aspm KO mice and experiment methods employed, including the ages or markers analyzed. 

Refer to Fig 1

4. The authors propose a reduction in Tbr1+ thickness (line 190, Fig 1H-K). This would benefit from determining differences in cell numbers instead of thickness, in order to assert whether neurogenesis is affected, and considering the effect on thickness is mild.

Importantly, this reduction is not consistent with later results (Fig.3) where birth dating experiments at E12.5 do not show deep layer differences between WT and cKO.

I would greatly appreciate your critical suggestions.

According to the suggestion, we revised Fig 1. 

　

Page 9, line 210

Fig 1. Embryonal brain of Aspm cKO

(A-C) Representative DAPI stained images of the embryonal cortices of the WT and Aspm cKO mice (NesCre;Aspmflox/flox) at E12.5, E14.5 and E16.5. The scale bars represent 500 μm. (A’-C’) Magnification of the boxed area in (A-C). The scale bars represent 50 μm. (D) Quantification of the thickness per unit area (A’-C’) (250 μm radial columns) *: P < 0.05 (E12.5: WT 151.1 ± 2.0 μm, cKO 155.1 ± 6.5 μm, P = 0.5922, E14.5: WT 238.0 ± 13.5 μm, cKO 226.9 ± 5.8 μm, P = 0.4913, E16.5: WT 482.4 ± 7.5 μm, cKO 427.1 ± 15.5 μm, P = 0.0325, Student’s t-test, values represent mean ± SEM, n = 3 mice per genotype, ns: not significant) (E) SOX2 and DAPI stained image of E16.5 mouse cerebral cortex. The two head arrows indicate the CP, VZ and cortical wall. (F) Quantification of cortical wall and the CP region were defined based on DAPI staining and that of VZ was defined based on SOX2 staining. The IZ was defined as the value obtained by subtracting the CP and VZ from cortical wall. (VZ, WT 85.54 ± 2.2 μm, cKO 70.59 ± 2.1 μm, P = 0.0104; IZ, WT　229.29 ± 4.9 μm, cKO 210.55 ± 9.3 μm, P = 0.1490; CP, WT 159.51 ± 6.4 μm, cKO 140.03 ± 9.0 μm, P = 0.0908, Student’s t-test, values represent mean ± SEM, n = 3 mice per genotype) (G-I) Representative images of the WT and Aspm cKO Satb2, Ctip2 and Tbr1, respectively. Scale bar represents 50 µm. (J-L) Quantification of the cell number of the upper layer marker Satb2, the middle layer maker Ctip2 and the deep layer marker Tbr1, respectively, per unit area G-I (250μm radial columns). (J: WT 314.0 ± 23.4 cells, cKO 266.7 ± 36.2 cells, P = 0.3336, K: WT 392.7 ± 19.3 cells, cKO 327.7 ± 26.8 cells, P = 0.1205, L: WT 416.0 ± 14.7 cells, cKO 377.3 ± 17.9 cells, P = 0.1702, Student’s t-test, values represent mean ± SEM, n = 3 mice per genotype, ns: not significant).

Page 9, line 188

First, we generated brain specific Aspm knockout mouse (NesCre;Aspmflox/flox, Aspm cKO) (S1 Fig) in order to examine whether the morphology of the embryonal cerebral cortex was affected by the loss of Aspm. Cortical thickness did not significantly change between the WT and cKO mice at E12.5 and E14.5 (E12.5, WT 151.1 ± 2.0 µm, cKO 155.1 ± 6.4 µm, P = 0.5922; E14.5, WT 238.0 ± 13.5 µm, cKO 227.0 ± 5.8 µm, P = 0.4913). However, a morphological difference was observed at E16.5, with a decrease in cortical thickness (WT 482.4 ± 7.5 µm, cKO 427.1 ± 15.5 µm, P = 0.0325) (Figs 1A-1D).

We further analyzed how the changes were related to the layer construction of the cerebral cortex at E16.5. First, there was a significant decrease in the thickness of the VZ in Aspm cKO (Figs 1C’, 1E and 1F, and S8 Fig). Although the thickness of IZ or CP showed a slight decrease, the significance did not reach the level of significance as compared to WT (Figs 1C’, 1E and 1F, and S6 Fig).

The CP is composed of six layers, and each layer consists of neurons with different birthdates. Therefore, we examined the CP in more detail using specific markers for each cortical layer. In the mouse brains lacking Aspm, the number of cells immunoreactive for the upper layer marker (Satb2-positive, II/III and IV layer), the middle layer marker (Ctip2-positive, layer V) and for the deep layer marker (Tbr1-positive, layer VI) showed a slight decrease, but no significant differences, when compared to those of the WT mice, (Figs 1G-1L), suggesting that a normal cytoarchitecture of the cerebral cortex was formed even with a loss of Aspm. 

These findings suggest that the thinning of the cortex in late cortical formation might be due to poor neurogenesis or an excess loss of neurons at all embryonal stages.

Refer to Fig3

5-1. In EdU/BrdU labeling experiments (Fig 3) it would be useful to also test for differences in number and distribution of EdU cells at a later timepoint (P5-10) when neurons have completed migration. The reduction in EdU cell numbers seen in the IZ but not CP, which suggests they are later born, would also be supported by Tbr1 co-staining. Authors should also consider adding an EdU pulse at E16.5 to solidly conclude which neurons are affected.

I would greatly appreciate your critical suggestions.

We agree with you that the reduction in EdU-labeled cell numbers seen in the IZ but not CP, which suggests they are later born neurons (later than E14.5). 

From the S7 Fig, around 90% of EdU-positive cells showed immunoreactivity for Satb2, showing that they might be later-born neurons

Since we could not generate new animals, we could not analyze EdU pulse study at E16.5.

5-2. Additionally, in Fig 3 E-F, no EdU + cells are shown in quantification in bins 1-2 (or VZ/SVZ), but they are clearly evident in the images. Distinguishing between EdU cells remaining in VZ or migrating into IZ (with short chase times), coupled with the use of specific markers could help determine whether there is changes in neurogenic divisions. For this point, authors should consider similarities and/or differences with published work.

I would greatly appreciate your important suggestions.

According to your suggestions, in terms of the birthdating assay at early to middle stage, we have confirmed co-immunostaining of EdU and specific neural maker such as Tbr1 and remade the Fig 3, which includes co-immunostaining for Tbr1, EdU and DAPI images in order to show that the EdU positive cells were postmitotic neurons in cortical plate at E14.5 (Figs 3A and B).

We described as follows: 

Page 12, line 263

Most of the cells, EdU-labeled at E12.5, were observed in the superficial area corresponding to the CP, and EdU-positive cells were found in postmitotic neurons showing Tbr1-immunoreactivity. The E12.5-labeled neurons showed no significant differences between the WT and cKO mice (Figs 3A and 3B).

Regarding the double staining ratio, we made supplemental figure (S7 Fig), which revealed that Tbr1/EdU was about 8.7% in WT and 9.9% in cKO, that of Ctip2/EdU was about 3.4% in WT and about 3.4% in cKO, and that of Satb2/EdU was about 76.6% in WT and 86.9% in cKO, respectively (S7 Fig). 

We added the description as follows:

Page 12, line 266.

In contrast, the number of cells labeled at E14.5 was significantly reduced, with most of them scattered from the IZ to the CP, in the cKO mice. There was a significant reduction in the number of EdU-labeled cells distributed in the IZ, but not in the VZ and CP (Figs 3C-3F). Although most of the EdU-labeled cells showed immunoreactivity for Satb2, late born neurons, there were no significant differences in ratio of Tbr1+/EdU+, Ctip2+/EdU+, and Satb2+/EdU+ cells in the cerebral wall between the WT and Aspm cKO brains (S7 Fig). 

These findings suggested that the loss of Aspm induced a reduction in the number of mid to late born neurons without any aberrant expression of cerebral layer-specific transcription factors in the developing cortex.

As suggested, we have remeasured for the number of EdU-positive cells and remade the Fig 3, which included quantitative results about VZ/SVZ (bin1/bin2). Furthermore, it was performed Edu+/Tbr1+ cells because neurons at the lower CP and subplate (SP) were Tbr1-positive and the border between CP and SP was distinguishable to indicate the cortical layer (CP, IZ and VZ/SVZ) and bin number (Bin1-10).

We also have remade the figure legend as follows: 

Page 12, line 276

Fig 3. Neurogenesis during murine corticogenesis. 

(A, C) Co-staining images of Tbr1 (green), EdU (red) and DAPI (blue) in the WT and Aspm cKO mice at E14.5 (A) and E16.5 (C) cortices two days after the EdU injections (E12.5 and E14.5, respectively). Scale bar: 50 µm. (B, D) The number of EdU-positive postmitotic cells per 250 µm. *P < 0.05 (B: WT 50.67 ± 3.5, cKO 54.3 ± 7.9, P = 0.6923, D: WT 228.7 ± 6.9, cKO 174.7 ± 11.4, P = 0.0154, Student’s t-test, values represent mean ± SEM, n = 3 mice per genotype) (E) Histogram of the EdU-positive cells, which were shown in C. (bins × genotype interaction F(8, 36) = 2.893 P = 0.0135; two-way ANOVA with Tukey’s multiple comparisons test; **P < 0.01, values represent mean ± SEM, n = 3 mice per genotype). (F) Comparison of the distribution of the EdU-positive cells among the cortical layers (cortical layers × genotype interaction F(2, 12) = 27.82 P < 0.0001; two-way ANOVA with Tukey’s multiple comparisons test; ****P < 0.0001, ns: not significant, values represent mean ± SEM, n = 3 mice per genotype)

Minor

Refer to S4 Fig

6． Examples of DNA damage in both Pax6+ and Pax6- cells are shown in Fig Suppl 4. Since numbers were not quantified, lines 314-315 should be revised. Quantifications of both cell type specific death and DNA damage would increase the robustness of the results.

I would greatly appreciate your critical suggestions.

According to your suggestions, we omitted ‘increase’ from the description.

Page 15, line 341.

Furthermore, the double immunofluorescence of γH2AX with the neural progenitor cell marker, Pax6, revealed that the cells with DNA damage were neural progenitor cells expressing Pax6 (S4 Fig).

7． It would be ideal to show the actual data points in the graphs together with mean +/- SEM.

I would greatly appreciate your critical suggestions.

We added actual data in figure legends.

Refer to Abstract

8． Please, rephrase “..observed cortical reduction in the late neurogenesis of murine cortical development” Abstract, line 27.

I would greatly appreciate your critical suggestions.

We rephrased the sentence as follows:

Page 1, line 27

In this study, we generated brain-specific Aspm knockout mice to evaluate the fetal brain phenotype and observed cortical reduction in the late stage of murine cortical development.

9． Line 59: mutations in ASPM are a cause (not consequence) of mitotic aberrations during neurogenesis.

I would greatly appreciate your critical suggestions.

We changed the sentence as follows:

Page 3, line 58. 

Therefore, mutations in the ASPM gene are a cause of mitotic aberrations during neurogenesis.

10． References for mouse lines are missing in Methods section.

I would greatly appreciate your critical suggestions.

We added in Materials and Methods as follows:

Page 4, Line 82.

In order to obtain NesCre;Aspmflox/+ mice, Nestin-cre (C57BL/6-Tg(Nes-Cre)1Kag, Center for Animal Resources and Development (CARD), CARD-ID 650) mice [16] were mated with Aspmflox/flox mice [9]. Brain specific Aspm knockout mice (NesCre;Aspmflox/flox, Aspm cKO) were generated by mating Aspmflox/flox mice and NesCre;Aspmflox/+ mice and detected by genotyping.

11． Please indicate ages in Figures 1 F-H, 1E, 3 A-F where they are missing so that is easily interpreted.

I would greatly appreciate your critical suggestions.

We added age in Fig 1F-H, and Fig 3A-F. 

12． Label the stainings close/within the panels in all figures. They are missing in Figs. 2, 3A-C, 5A-F and Fig.S2 A-F.

I would greatly appreciate your critical suggestions.

We labeled the staining in Figs. 2, 3A-C, 5A-F and Fig.S2 A-F.

Fig 2

Fig 3

Fig 5

S2 Fig 

13． Line 339 should be modified. The authors show that there is no excessive or decreased proliferation but can’t exclude that there are other ways of aberrant division such as altered ratio of self-renewal vs. neurogenic divisions.

I would greatly appreciate your critical suggestions.

We modified the description as follows:

Page 17, line 374

These findings suggest that the microcephaly associated with MCPH5 might be aggravated by increased apoptosis in NPCs and postmitotic neurons, but not by aberrant proliferation of NPCs during cortical development, although we can’t exclude that there are other ways of aberrant division such as altered ratio of self-renewal vs. neurogenic divisions.

14． Please avoid the use of “cortical areas” as it may get confused with functional areas of the cortex (line 270). I suggest to remove the word “layers” (line 32, abstract)

I would greatly appreciate your critical suggestions.

We removed the ‘areas” as follows:

Page 1, line 30.

On the other hand, the knockout mice showed a constant increase in apoptosis in the cerebral cortex from the early, through the late stages of cortical development.

Page 13, line 295

The numerical density of the TUNEL-positive cells in the developing cortex (TUNEL index) decreased as the embryonal stage proceeded (Figs 4H and 4I).

15． I think it is more common to use embryonic instead of embryonal (176, 190 and others)

I would greatly appreciate your critical suggestions.

We changed “embryonic” to “embryonal” throughout the manuscript.

---

## [Decision Letter · Decision Letter 1]

11 Aug 2023

PONE-D-23-02973R1Loss of Aspm causes increased apoptosis of developing neuronal cells during mouse cerebral corticogenesisPLOS ONE

Dear Dr. Itoh,

Thank you for submitting  the revised version of your manuscript to PLOS ONE. After careful consideration, we feel that you still need to make some corrections before your paper is ready for publication. Therefore, we invite you to submit a revised version of the manuscript that addresses the points raised during this second round of review process. Important to notice that you do not need to perform new experiments, but have to make corrections regarding data presentation (use of repetitive images among other points raised) and interpretation of some results.

Please submit your revised manuscript by Sep 25 2023 11:59PM. If you will need more time than this to complete your revisions, please reply to this message or contact the journal office at plosone@plos.org. Please include the following items when submitting your revised manuscript:A rebuttal letter that responds to each point raised by the academic editor and reviewer(s). You should upload this letter as a separate file labeled 'Response to Reviewers'.A marked-up copy of your manuscript that highlights changes made to the original version. You should upload this as a separate file labeled 'Revised Manuscript with Track Changes'.An unmarked version of your revised paper without tracked changes. You should upload this as a separate file labeled 'Manuscript'.If applicable, we recommend that you deposit your laboratory protocols in protocols.io to enhance the reproducibility of your results. Protocols.io assigns your protocol its own identifier (DOI) so that it can be cited independently in the future. For instructions see: https://journals.plos.org/plosone/s/submission-guidelines#loc-laboratory-protocols. Additionally, PLOS ONE offers an option for publishing peer-reviewed Lab Protocol articles, which describe protocols hosted on protocols.io. Read more information on sharing protocols at https://plos.org/protocols?utm_medium=editorial-email&utm_source=authorletters&utm_campaign=protocols.

We look forward to receiving your revised manuscript.

Kind regards,

Carlos Oliva, PhD

Academic Editor

PLOS ONE

Journal Requirements:

Reviewers' comments:

Reviewer's Responses to Questions

**Comments to the Author**

1. If the authors have adequately addressed your comments raised in a previous round of review and you feel that this manuscript is now acceptable for publication, you may indicate that here to bypass the “Comments to the Author” section, enter your conflict of interest statement in the “Confidential to Editor” section, and submit your "Accept" recommendation.

Reviewer #1: (No Response)

Reviewer #2: All comments have been addressed

2. Is the manuscript technically sound, and do the data support the conclusions?

Reviewer #1: Partly

Reviewer #2: Yes

3. Has the statistical analysis been performed appropriately and rigorously? 

Reviewer #1: Yes

Reviewer #2: Yes

4. Have the authors made all data underlying the findings in their manuscript fully available?

Reviewer #1: Yes

Reviewer #2: Yes

5. Is the manuscript presented in an intelligible fashion and written in standard English?

Reviewer #1: Yes

Reviewer #2: Yes

6. Review Comments to the Author

Reviewer #1: The reviewer appreciates that the authors have improved the manuscript, including most of our comments and suggestions. However, there are still important issues to be addressed.

The main issue is that the authors misinterpret or overstate their conclusions. Positively, they found reductions in total cortical (Fig. 1D) and VZ thickness (Fig. 1F), probably due to NPC apoptosis (Fig. 4), as they did not find changes in proliferation (Fig. 2) or neurogenesis (Fig. 1G-L). Although the authors showed a decrease in EdU-positive cells (Fig. 3D), these are mostly located in the VZ/SVZ or IZ areas, arguing that the main cell population affected by the loss of ASPM are NPCs rather than neurons. Indeed, in Figure 4, they found an increase in TUNEL+ cells exclusively at the VZ/SVZ.

Thus, I strongly suggest that the authors adjust their conclusions to the data presented in their manuscript and to the limitations of the experimental approach, avoiding overstating their conclusions.

Other comments:

1. "Line 207-208: These findings suggest that the thinning of the cortex in late cortical formation might be due to poor neurogenesis or an excess loss of neurons at all embryonal stages."

The data presented here does not allow the authors to conclude this as the only cortical layer affected is the ventricle zone, suggesting that ASPM is important for NPC maintenance, not neurogenesis or differentiation from NPCs to neurons. Indeed, the IZ or CP areas are unaffected by the ASPM loss.

2. "Lines 232-238: Next, we conducted assays to determine whether or not the thinning of the deep cortical layer caused by Aspm deficiency is related to changes in the neural progenitor cell proliferation. In order to identify specific neural progenitors, apical progenitor cells (APs) were defined as the phospho-histone H3 (pHH3) positive cells located beneath the surface of the lateral ventricle, and basal progenitor cells (BPs) were defined as the pHH3 positive cells localized in the basal region of the ventricular zone (Fig 2) [17]. Immunofluorescence studies for pHH3, which is the mitotic 237 marker, revealed no significant differences in the abundance of APs and BPs between the WT and Aspm cKO mice (Figs 2A and 2B)."

First, the authors do not show the thinning of the deep cortical layer. They showed the thinning of total cortical thickness and also of the VZ only. These data also argue that ASPM is important for NPC maintenance rather than neurogenesis.

Then, using the experimental approach presented, the authors can not assess the abundance of APs or BPs, as they did not use specific cell markers and pHH3 only labels cells during mitosis, which are quite few compared with the total AP or BP population. They can infer whether there are changes in NPCs positive for pHH3 only using their spatial location but not changes in NPC pools.

3. "Lines 263-265: Most of the cells, EdU-labeled at E12.5, were observed in the superficial area corresponding to the CP, and EdU-positive cells were found in postmitotic neurons showing Tbr1-immunoreactivity. The E12.5-labeled neurons showed no significant differences between the WT and cKO mice (Figs 3A and 3B). In contrast, the number of cells labeled at E14.5 was significantly reduced, with most of them scattered from the IZ to the CP, in the cKO mice."

It was impossible to check this statement as there is no graph showing changes in VZ/IZ/CP distribution in EdU-labeled cells at E12.5.

Also, EdU-labeled cells at E14.5 data showed that EdU+ cells are preferentially located in the IZ, where it is possible to find Tbr2+ intermediate progenitors. Although several cells are located at the CP, there are also located at the VZ. Then, it is not correct to assume that they are postmitotic neurons.

Reviewer #2: In this revised version of manuscript, the authors have addressed the major points of my previous review. The authors have included additional data which have improved the presentation.

I have a few observations that should be addressed in the final version

1. Some images are repetitively used in different figures of the manuscript.

The photos illustrating the E16.5 cortex of wild type and Aspm-cKOs shown in Fig.1C, in Fig.3C, in Fig.S6C and in Fig.S7 A,D are exactly the same pictures. In some instances, they are labeled as DAPI, in others as YOYO1.

Please replace the images to avoid repetitive use of the same representative photos.

2. Line 232-233. Revise the phrase “….the thinning of the deep cortical layer caused by Aspm deficiency ….”. This is contradictory with the results in Fig.1 I,L that shows that deep layer Tbr1+ is not significantly different in mutants.

3. The Suppl. Figures should be renumbered according to the order they are first mentioned in the text.

4. I suggest to use the term “embryonic”, instead of the term “embryonal”. It looks that my comment in the previous review was not understood correctly.

7. PLOS authors have the option to publish the peer review history of their article (what does this mean?). If published, this will include your full peer review and any attached files.

Reviewer #1: No

Reviewer #2: No

---

## [Author Response · Author response to Decision Letter 1]

29 Aug 2023

Reviewer #1: The reviewer appreciates that the authors have improved the manuscript, including most of our comments and suggestions. However, there are still important issues to be addressed.

The main issue is that the authors misinterpret or overstate their conclusions. Positively, they found reductions in total cortical (Fig. 1D) and VZ thickness (Fig. 1F), probably due to NPC apoptosis (Fig. 4), as they did not find changes in proliferation (Fig. 2) or neurogenesis (Fig. 1G-L). Although the authors showed a decrease in EdU-positive cells (Fig. 3D), these are mostly located in the VZ/SVZ or IZ areas, arguing that the main cell population affected by the loss of ASPM are NPCs rather than neurons. Indeed, in Figure 4, they found an increase in TUNEL+ cells exclusively at the VZ/SVZ.

Thus, I strongly suggest that the authors adjust their conclusions to the data presented in their manuscript and to the limitations of the experimental approach, avoiding overstating their conclusions.

I would greatly appreciate your critical comments. Because of the limitations of the experimental approach, it is difficult to conclude that loss of Aspm mainly affected NPCs or postmitotic immature neurons. According to your suggestions, we revised the descriptions as follows.

Page 9, line 207

These findings suggest that Aspm is important for NPC maintenance, not for neurogenesis or differentiation from NPCs to neurons at all embryonic stages.

Page 11, line 245

These results suggest that loss of Aspm does not affect the proliferation of the NPCs, but might affect NPCs pool during mouse neurogenesis. 

Page 12, line 275

These findings suggested that the loss of Aspm induced a reduction in the number of mid to late born neurons without any aberrant expression of cerebral layer-specific transcription factors in the developing cortex, which might be due to affected NPCs pool. 

Page 21, line 471

These findings suggest that Aspm deficiency can induce apoptotic loss of NPCs and immature neurons, leading to decreased NSCs pool and neuronal loss and ultimately to phenotypic change (i.e., cortical thinning) as found in MCPH.

Other comments:

1. "Line 207-208: These findings suggest that the thinning of the cortex in late cortical formation might be due to poor neurogenesis or an excess loss of neurons at all embryonal stages."

The data presented here does not allow the authors to conclude this as the only cortical layer affected is the ventricle zone, suggesting that ASPM is important for NPC maintenance, not neurogenesis or differentiation from NPCs to neurons. Indeed, the IZ or CP areas are unaffected by the ASPM loss.

I would greatly appreciate your critical comments. According to your suggestions, we revised the descriptions as follows.

Page 9, line 207

These findings suggest that Aspm is important for NPC maintenance, not for neurogenesis or differentiation from NPCs to neurons at all embryonic stages.

2. "Lines 232-238: Next, we conducted assays to determine whether or not the thinning of the deep cortical layer caused by Aspm deficiency is related to changes in the neural progenitor cell proliferation. In order to identify specific neural progenitors, apical progenitor cells (APs) were defined as the phospho-histone H3 (pHH3) positive cells located beneath the surface of the lateral ventricle, and basal progenitor cells (BPs) were defined as the pHH3 positive cells localized in the basal region of the ventricular zone (Fig 2) [17]. Immunofluorescence studies for pHH3, which is the mitotic marker, revealed no significant differences in the abundance of APs and BPs between the WT and Aspm cKO mice (Figs 2A and 2B)."

First, the authors do not show the thinning of the deep cortical layer. They showed the thinning of total cortical thickness and also of the VZ only. These data also argue that ASPM is important for NPC maintenance rather than neurogenesis.

Then, using the experimental approach presented, the authors can not assess the abundance of APs or BPs, as they did not use specific cell markers and pHH3 only labels cells during mitosis, which are quite few compared with the total AP or BP population. They can infer whether there are changes in NPCs positive for pHH3 only using their spatial location but not changes in NPC pools.

I would greatly appreciate your critical comments. According to your suggestions, we revised the descriptions as follows.

Page 11, line 245

These results suggest that loss of Aspm does not affect the proliferation of the NPCs, but might affect NPCs pool during mouse neurogenesis. 

3. "Lines 263-265: Most of the cells, EdU-labeled at E12.5, were observed in the superficial area corresponding to the CP, and EdU-positive cells were found in postmitotic neurons showing Tbr1-immunoreactivity. The E12.5-labeled neurons showed no significant differences between the WT and cKO mice (Figs 3A and 3B). In contrast, the number of cells labeled at E14.5 was significantly reduced, with most of them scattered from the IZ to the CP, in the cKO mice."

It was impossible to check this statement as there is no graph showing changes in VZ/IZ/CP distribution in EdU-labeled cells at E12.5.

I would greatly appreciate your critical comments. According to your suggestions, we assessed VZ/IZ/CP distribution in EdU-labeled cells at E12.5.

Page 12, line 265

The number of EdU-labeled cells with Tbr1-immunoreactivity in the CP, as well as the number of EdU-labeled cells in the IZ and VZ showed no significant difference between the WT and Aspm cKO brains at E12.5 (Fig 3C).

Figure 3 was revised.

Fig 3. Neurogenesis during murine corticogenesis

(A, D) Co-staining images of Tbr1 (green), EdU (red) and YOYO1 (blue) in the WT and Aspm cKO mice at E14.5 (A) and E16.5 (D) cortices two days after the EdU injections (E12.5 and E14.5, respectively). Scale bar: 50 µm. (B, E) The number of EdU-positive cells per 250 µm. *P < 0.05 (B: WT 162.0 ± 6.1, cKO 169.7 ± 11.6, P = 0.5867, E: WT 228.7 ± 6.9, cKO 174.7 ± 11.4, P = 0.0154, Student’s t-test, values represent mean ± SEM, n = 3 mice per genotype) (C) Comparison of the distribution of the EdU-positive cells among the cortical layers (cortical layers × genotype interaction F(2, 12) = 0.3488 P < 0.7125; two-way ANOVA with Tukey’s multiple comparisons test; ns: not significant, values represent mean ± SEM, n = 3 mice per genotype). (F) Histogram of the EdU-positive cells, which were shown in C. (bins × genotype interaction F(8, 36) = 2.893 P = 0.0135; two-way ANOVA with Tukey’s multiple comparisons test; **P < 0.01, values represent mean ± SEM, n = 3 mice per genotype). (G) Comparison of the distribution of the EdU-positive cells among the cortical layers (cortical layers × genotype interaction F(2, 12) = 27.82 P < 0.0001; two-way ANOVA with Tukey’s multiple comparisons test; ****P < 0.0001, ns: not significant, values represent mean ± SEM, n = 3 mice per genotype).

Also, EdU-labeled cells at E14.5 data showed that EdU+ cells are preferentially located in the IZ, where it is possible to find Tbr2+ intermediate progenitors. Although several cells are located at the CP, there are also located at the VZ. Then, it is not correct to assume that they are postmitotic neurons.

I would greatly appreciate your critical comments. According to your suggestions, we revised the description as follows:

Page 12, line 267, 275

In contrast, the number of cells labeled at E14.5 was significantly reduced, with most of them located in the VZ/SVZ or IZ areas in the cKO mice (Figs 3D and 3E), arguing that the main cell population affected by the loss of ASPM were NPCs rather than neurons.

Page 12, line 275

These findings suggested that the loss of Aspm induced a reduction in the number of mid to late born neurons without any aberrant expression of cerebral layer-specific transcription factors in the developing cortex, which might be due to affected NPCs pool. 

 

Reviewer #2: In this revised version of manuscript, the authors have addressed the major points of my previous review. The authors have included additional data which have improved the presentation.

I have a few observations that should be addressed in the final version

1. Some images are repetitively used in different figures of the manuscript.

The photos illustrating the E16.5 cortex of wild type and Aspm-cKOs shown in Fig.1C, in Fig.3C, in Fig.S6C and in Fig.S7 A,D are exactly the same pictures. In some instances, they are labeled as DAPI, in others as YOYO1.

Please replace the images to avoid repetitive use of the same representative photos.

I would greatly appreciate your critical comments. According to your suggestions, we revised the figures.

Fig 1C: images of both WT and cKO were replaced.

Fig 1A’,B’, and C’: DAPI was changed to YOYO.

Fig 3 was revised:

DAPI was changed to YOYO.

Fig 3D (previous Fig 3C): images of both WT and cKO were replaced.

Fig 3B was changed and Fig 3C was added, and Fig numbering was changed due to insertion of a new data (Fig 3C)

Fig S6C was not changed.

S7 Fig A and D were replaced. The labels for EdU and Tbr1 were reversed and have been corrected.

2. Line 232-233. Revise the phrase “….the thinning of the deep cortical layer caused by Aspm deficiency ….”. This is contradictory with the results in Fig.1 I,L that shows that deep layer Tbr1+ is not significantly different in mutants.

I would greatly appreciate your critical comments. According to your suggestions, we revised as follows:

Page 10, line 232

Next, we conducted assays to determine whether or not the thinning of total cortical thickness and VZ thickness caused by Aspm deficiency is related to changes in the neural progenitor cell proliferation.

3. The Suppl. Figures should be renumbered according to the order they are first mentioned in the text.

I would greatly appreciate your important comments. According to your comments, we renumbered the Suppl. Figures according to the order.

S5 Fig was changed to S1 Fig.

S1 Fig was changed to S2 Fig.

S8 Fig was changed to S3 Fig.

S6 Fig was changed to S4 Fig.

S7 Fig was changed to S5 Fig.

S2 Fig was changed to S6 Fig.

S3 Fig was changed to S7 Fig.

S4 Fig was changed to S8 Fig.

4. I suggest to use the term “embryonic”, instead of the term “embryonal”. It looks that my comment in the previous review was not understood correctly.

I would greatly appreciate your important comments. According to your suggestions, we revised to “embryonic”, instead of the term “embryonal”.

---

## [Decision Letter · Decision Letter 2]

7 Nov 2023

PONE-D-23-02973R2Loss of Aspm causes increased apoptosis of developing neuronal cells during mouse cerebral corticogenesisPLOS ONE

Dear Dr. Itoh,

Thank you for submitting your manuscript to PLOS ONE. After this round of revision, both reviewers agreed that most of the previous comments were addressed, however there are still minor critics to consider before publication. Therefore, we invite you to submit a revised version of the manuscript that addresses the points raised during the review process.

 In particular, one of the reviewers asked to revise some of the conclusions of the manuscript, based on the data presented. Since, it is possible that this will be the last round of revision, I recommend to do a final check of typos in your manuscript. Please submit your revised manuscript by Dec 22 2023 11:59PM. If you will need more time than this to complete your revisions, please reply to this message or contact the journal office at plosone@plos.org. Please include the following items when submitting your revised manuscript:A rebuttal letter that responds to each point raised by the academic editor and reviewer(s). You should upload this letter as a separate file labeled 'Response to Reviewers'.A marked-up copy of your manuscript that highlights changes made to the original version. You should upload this as a separate file labeled 'Revised Manuscript with Track Changes'.An unmarked version of your revised paper without tracked changes. You should upload this as a separate file labeled 'Manuscript'.If applicable, we recommend that you deposit your laboratory protocols in protocols.io to enhance the reproducibility of your results. Protocols.io assigns your protocol its own identifier (DOI) so that it can be cited independently in the future. For instructions see: https://journals.plos.org/plosone/s/submission-guidelines#loc-laboratory-protocols. Additionally, PLOS ONE offers an option for publishing peer-reviewed Lab Protocol articles, which describe protocols hosted on protocols.io. Read more information on sharing protocols at https://plos.org/protocols?utm_medium=editorial-email&utm_source=authorletters&utm_campaign=protocols.

We look forward to receiving your revised manuscript.

Kind regards,

Carlos Oliva, PhD

Academic Editor

PLOS ONE

Journal Requirements:

Reviewers' comments:

Reviewer's Responses to Questions

**Comments to the Author**

1. If the authors have adequately addressed your comments raised in a previous round of review and you feel that this manuscript is now acceptable for publication, you may indicate that here to bypass the “Comments to the Author” section, enter your conflict of interest statement in the “Confidential to Editor” section, and submit your "Accept" recommendation.

Reviewer #1: (No Response)

Reviewer #2: All comments have been addressed

2. Is the manuscript technically sound, and do the data support the conclusions?

Reviewer #1: Partly

Reviewer #2: Yes

3. Has the statistical analysis been performed appropriately and rigorously? 

Reviewer #1: Yes

Reviewer #2: Yes

4. Have the authors made all data underlying the findings in their manuscript fully available?

Reviewer #1: (No Response)

Reviewer #2: Yes

5. Is the manuscript presented in an intelligible fashion and written in standard English?

Reviewer #1: Yes

Reviewer #2: Yes

6. Review Comments to the Author

Reviewer #1: The authors have significantly improved this manuscript, and their conclusions about NPC apoptosis are well supported by their data. However, the data about neuronal apoptosis is still weak. Thus, I have some minor suggestions to make their conclusions more accurate:

Line 1: replace neuronal by neural

Line 32: eliminate "and postmitotic neurons"

Line 34: eliminate "neuronal"

Line 73: eliminate "and neurons"

Line 295: eliminate "and postmitotic neurons"

Reviewer #2: The comments made to the previous versions of the manuscript were addressed in the revised R2 version

7. PLOS authors have the option to publish the peer review history of their article (what does this mean?). If published, this will include your full peer review and any attached files.

Reviewer #1: No

Reviewer #2: No

---

## [Author Response · Author response to Decision Letter 2]

7 Nov 2023

I would greatly appreciate the reviewer #1’s critical comments, and I revised the manuscript: PONE-D-23-02973R2 according to the reviewer’s comments.

Reviewer #1: The authors have significantly improved this manuscript, and their conclusions about NPC apoptosis are well supported by their data. However, the data about neuronal apoptosis is still weak. Thus, I have some minor suggestions to make their conclusions more accurate:

Line 1: replace neuronal by neural

I replaced neuronal by neural.

Line 32: eliminate "and postmitotic neurons"

I eliminated "and postmitotic neurons"

Line 34: eliminate "neuronal"

I am sorry but I could not find "neuronal" in Line 34.

Line 73: eliminate "and neurons"

I eliminated "and neurons"

Line 295: eliminate "and postmitotic neurons"

I eliminated "and postmitotic neurons"

---

## [Editor Report · Decision Letter 3]

13 Nov 2023

Loss of Aspm causes increased apoptosis of developing neural cells during mouse cerebral corticogenesis

PONE-D-23-02973R3

Dear Dr. Itoh,

We’re pleased to inform you that your manuscript has been judged scientifically suitable for publication and will be formally accepted for publication once it meets all outstanding technical requirements.

Kind regards,

Carlos Oliva, PhD

Academic Editor

PLOS ONE
---

## [Editor Report · Acceptance letter]

17 Nov 2023

PONE-D-23-02973R3 

Loss of Aspm causes increased apoptosis of developing neural cells during mouse cerebral corticogenesis 

Dear Dr. Itoh:

I'm pleased to inform you that your manuscript has been deemed suitable for publication in PLOS ONE. Congratulations! Your manuscript is now with our production department. 

Kind regards, 

on behalf of

Dr. Carlos Oliva 

Academic Editor

PLOS ONE